# Joker: Joint Optimization Framework for Lightweight Kernel Machines

**Junhong Zhang** [1]   **Zhihui Lai** [1 2]

## Abstract

Kernel methods are powerful tools for nonlinear learning with well-established theory. The scalability issue has been their long-standing challenge. Despite the existing success, there are two limitations in large-scale kernel methods: (i) The memory overhead is too high for users to afford; (ii) existing efforts mainly focus on kernel ridge regression (KRR), while other models lack study. In this paper, we propose Joker, a joint optimization framework for diverse kernel models, including KRR, logistic regression, and support vector machines. We design a dual block coordinate descent method with trust region (DBCD-TR) and adopt kernel approximation with randomized features, leading to low memory costs and high efficiency in large-scale learning. Experiments show that Joker saves up to 90% memory but achieves comparable training time and performance (or even better) than the state-of-the-art methods.

## 1. Introduction

Kernel methods, a formidable paradigm in nonlinear learning, stand alongside deep learning as a prominent approach in the machine learning landscape. Recently, theoretical progress that connects the modern deep neural network with the kernel machines has been made (Jacot et al., 2018; Geifman et al., 2020; Chen & Xu, 2020), highlighting the great potential of kernel methods. In the big data era, kernel methods' scalability has become a concern. Kernel ridge regression (KRR) is the simplest kernel model. However, it is associated with a linear system potentially requiring $O(n^3)$ operations, where $n$ is the size of the training set. Another instance is the kernel support vector machine (SVM) (Cortes & Vapnik, 1995). Despite its prevalence, it also faces scal-

ability challenges with large datasets due to the induced large-scale quadratic programming (Wen et al., 2018).

Numerous researchers have dedicated their efforts to tackling the scalability issues. Nyström method (Williams & Seeger, 2000) and randomized features (Rahimi & Recht, 2007) are two representative techniques to reduce the computation of kernel methods. Nevertheless, researchers preferred the former one (Yang et al., 2012). Based on Nyström method, Rudi et al. (2017) proposed a KRR model, Falkon, for large-scale data. Its variants (Rudi et al., 2018; Meanti et al., 2020) still hold the state-of-the-art performance and efficiency for large-scale kernel methods. However, the outstanding performance is priced at the expense of high memory usage. Meanti et al. (2020) suggests that a good performance of Falkon-based methods necessitates a large number of Nyström centers. Yet, they require $O(M^2)$ memory to store the preconditioner for acceleration, where $M$ is the number of Nyström centers. This leads to a dilemma between memory consumption and performance. To perform Falkon with $M = 1.2 \times 10^5$ on the HIGGS dataset as in (Meanti et al., 2020) using single precision, one needs at least 55GB of memory, which is unaffordable for most users. In other words, memory resources become a bottleneck for large kernel machines (Abedsoltan et al., 2023).

We also note that the existing large-scale kernel methods mainly focus on KRR while other models lack study. For classification models, to our knowledge, the recent references about large-scale kernel logistic regression (KLR) and SVM [1], which are two classifiers commonly used in practice, are relatively sparse. (Marteau-Ferey et al., 2019) and (Wen et al., 2018) are the two latest influential works about KLR and SVM, respectively. However, they still meet the bottlenecks of either high memory or time consumption.

In short, this paper aims to address two issues of the large-scale kernel methods: *expensive memory requirement* and *limited model diversity*. We propose a joint optimization framework for kernel machines, Joker, which significantly reduces memory footprint while keeping comparable performance and efficiency to the state of the art, and covers a broad range of models with a unified optimization scheme. The contributions, some backgrounds, and a detailed review

---

[1]College of Computer Science and Software Engineering, Shenzhen University, Shenzhen, 518060, China [2]Guangdong Provincial Key Laboratory of Intelligent Information Processing, Shenzhen, 518060, China. Correspondence to: Zhihui Lai <lai_zhi_hui@163.com>.

*Proceedings of the 42$^{nd}$ International Conference on Machine Learning*, Vancouver, Canada. PMLR 267, 2025. Copyright 2025 by the author(s).

---

[1]SVM includes SVC and SVR, where "C" and "R" stand for classification and regression, respectively. Here we refer to SVC.

*Table 1.* Comparison of large-scale kernel methods on the HIGGS dataset. The "hybrid" means Joker supports both exact and inexact kernel models.

| Method | Type | Memory | Time | Models |
|---|---|---|---|---|
| Falkon | inexact | > 50GB | < 1 hour | KRR |
| LogFalkon | inexact | > 50GB | < 1 hour | KLR |
| EigenPro3 | inexact | ≈ 7GB | > 15 hours | KRR |
| LIBSVM | exact | < 2GB | > 1 week | SVC, SVR |
| ThunderSVM | exact | ≈ 8GB | > 1 week | SVC, SVR |
| Joker (ours) | hybrid | ≈ 2GB | ≈ 1 hour | see Table 2 |

of related work are presented in the rest of this section.

## 1.1. Contributions

Our contributions are summarized as three breakthroughs:

- **Unified scheme**: We develop Joker, a joint optimization framework for diverse kernel models beyond KRR, presenting a general scheme to support a wide range of large-scale kernel machines.

- **Low consumption**: Joker is lightweight. We propose a novel solver, dual block coordinate descent with trust region (DBCD-TR). Owing to its low space and time complexity, the hardware requirement of large-scale learning is significantly reduced.

- **Superior performance**: We implemented KRR, KLR, SVM, etc. based on Joker, and conducted experiments with a *single RTX 3080 (10GB)*. It shows that Joker achieves state-of-the-art performance with a low memory budget and moderate training time.

## 1.2. Preliminaries

This paper focuses on supervised learning problems. Let $\mathcal{X}$ be a compact sample space, and $\mathcal{Y}$ be a label set. The training dataset $\{(\boldsymbol{x}_i, y_i)\}_{i=1}^n$ includes $n$ samples, where $\boldsymbol{x}_i \in \mathcal{X}$ is the feature vector of the $i$-th sample and $y_i \in \mathcal{Y}$ is its label. For a function $f$, its domain is dom $f = \{\boldsymbol{x} \in \mathcal{X} : f(\boldsymbol{x}) < \infty\}$, and its gradient and Hessian at $\boldsymbol{x}$ are $\nabla f(\boldsymbol{x})$ and $\nabla^2 f(\boldsymbol{x})$, respectively. The Fenchel conjugate of $f$ is $f^*(\boldsymbol{y}) := \sup_{\boldsymbol{x} \in \text{dom } f} \langle \boldsymbol{y}, \boldsymbol{x} \rangle - f(\boldsymbol{x})$. The infimal convolution of $f$ and $g$ is $(f \square g)(\boldsymbol{u}) := \inf_{\boldsymbol{u}} f(\boldsymbol{u}) + g(\boldsymbol{x} - \boldsymbol{u})$. Let $[n] := \{1, 2, \cdots, n\}$. For two vectors $\boldsymbol{a}, \boldsymbol{b} \in \mathbb{R}^n$, $\boldsymbol{a} \le \boldsymbol{b}$ means $a_i \le b_i$ for all $i \in [n]$. Notation $\langle \boldsymbol{a}, \boldsymbol{b} \rangle$ represents the inner product between $\boldsymbol{a}$ and $\boldsymbol{b}$.

A Mercer's kernel $\mathcal{K}(\cdot, \cdot) : \mathcal{X} \times \mathcal{X} \mapsto \mathbb{R}$ uniquely induces a reproducing kernel Hilbert space (RKHS) $\mathcal{H}$ with the endowed inner product $\langle \cdot, \cdot \rangle_{\mathcal{H}}$, such that $\mathcal{K}(\boldsymbol{x}, \cdot) \in \mathcal{H}$ for all $\boldsymbol{x} \in \mathcal{X}$, and $\langle f, \mathcal{K}(\cdot, \boldsymbol{x}) \rangle_{\mathcal{H}} = f(\boldsymbol{x})$ for all $f \in \mathcal{H}$ and $\boldsymbol{x} \in \mathcal{X}$ (Schölkopf & Smola, 1998). We denote $\boldsymbol{K} \in \mathbb{R}^{n \times n}$ the kernel matrix with $K_{ij} = \mathcal{K}(\boldsymbol{x}_i, \boldsymbol{x}_j)$, where $i, j \in [n]$.

Let $\mathcal{I}, \mathcal{J} \subseteq [n]$ be index sets, and $\boldsymbol{K}_{\mathcal{I}, \mathcal{J}}$ be the submatrix of $\boldsymbol{K}$ with rows indexed by $\mathcal{I}$ and columns indexed by $\mathcal{J}$. We use $\boldsymbol{K}_{\mathcal{I}, :} := \boldsymbol{K}_{\mathcal{I}, [n]}$ and $\boldsymbol{K}_{:, \mathcal{J}} := \boldsymbol{K}_{[n], \mathcal{J}}$ for short. Let $\ell(\cdot, \cdot)$ be a proper and convex loss function. A generic kernel machine can be written as:

$$\min_{\boldsymbol{\theta} \in \mathcal{H}} \frac{\lambda}{2} \|\boldsymbol{\theta}\|_{\mathcal{H}}^2 + \sum_{i=1}^n \ell\left(y_i, \langle \boldsymbol{\theta}, \boldsymbol{\varphi}(\boldsymbol{x}_i)_{\mathcal{H}}\rangle\right), \qquad (1)$$

where $\boldsymbol{\varphi}(\boldsymbol{x})$ is a nonlinear map satisfying $\mathcal{K}(\boldsymbol{x}, \boldsymbol{x}') = \langle \boldsymbol{\varphi}(\boldsymbol{x}), \boldsymbol{\varphi}(\boldsymbol{x}') \rangle_{\mathcal{H}}$. The Representer Theorem (Schölkopf & Smola, 1998) states that the optimal solution $\boldsymbol{\theta}^\star$ satisfies $\boldsymbol{\theta}^\star = \sum_{i=1}^n \alpha_i^\star \boldsymbol{\varphi}(\boldsymbol{x}_i)$, for some $\boldsymbol{\alpha}^\star \in \mathbb{R}^n$. ***Exact models*** seeks such $\boldsymbol{\alpha}^\star$ without kernel approximation. For example, KRR uses $\ell(y, \hat{y}) = (y - \hat{y})^2 / 2$ and corresponds to a closed-form solution $\boldsymbol{\alpha}^\star = (\boldsymbol{K} + \lambda \boldsymbol{I})^{-1} \boldsymbol{y}$, however, which is computationally expensive for large-scale data. ***Inexact models*** approximate the kernel functions with less computational cost. For example, the Nyström-based method utilizes approximation $\boldsymbol{K} \approx \boldsymbol{K}_{:, \mathcal{I}} \boldsymbol{K}_{\mathcal{I}, \mathcal{I}}^{-1} \boldsymbol{K}_{\mathcal{I}, :}$ with a specific index set $\mathcal{I} \subset [n]$ satisfying $|\mathcal{I}| = M$ and $M \ll n$. It equivalent to restricting $\boldsymbol{\theta}$ in a subspace: $\boldsymbol{\theta} = \sum_{i \in \mathcal{I}} \alpha_i \boldsymbol{\varphi}(\boldsymbol{x}_i)$, making dimension of parameters largely reduced.

## 1.3. Related work

### 1.3.1. LARGE-SCALE KERNEL MACHINES

We start from KRR. Ma & Belkin (2017) proposed Eigen-Pro to solve the exact KRR using preconditioned gradient descent. Despite its success, its computational cost is still prohibitive. The Nyström method has become a dominant technique for inexact kernel models. Among them, Falkon (Rudi et al., 2017) leverages the conjugate gradient descent with a Cholesky-based preconditioner, which made significant progress on large-scale KRR with remarkable performance. The Falkon-based method (Rudi et al., 2018; Marteau-Ferey et al., 2019) usually requires $O(M^2)$ memory to store the Cholesky factor, where $M$ is the number of Nyström centers, resulting in memory limitation in practice. To this end, EigenPro3 avoids high space complexity by projected gradient descent (Abedsoltan et al., 2023), however, compromising with high time complexity per iteration. Hence, Abedsoltan et al. (2024) proposed a delayed projection technique to improve its efficiency. Finally, despite the relevance of KRR, we do not elaborate on works about Gaussian Process (de G. Matthews et al., 2017; Gardner et al., 2018) in this paper as the techniques are quite different.

Another focus of this paper lies in kernel models for classification. ThunderSVM implements the kernel SVM accelerated with GPU for large-scale data (Wen et al., 2018), emerging as a competent alternative to LIBSVM (Chang & Lin, 2011). However, their solver, Sequential Minimization Optimization (SMO) (Platt, 1998), becomes out of date for coping with big data in modern applications, causing high

*Table 2.* Fenchel conjugate associated with the losses. $\mathsf{bEnt}(x) := x \log x + (1-x) \log(1-x)$ denotes binary entropy, and $0 \log 0 := 0$. In the $L_p$-regression, we have $p, q > 1$ and $p^{-1} + q^{-1} = 1$.

| Task | Model | Loss functions | $\mathcal{Y}$ | $\xi_y(u)$ | $\xi_y^*(v)$ | dom $\xi_y^*$ |
|---|---|---|---|---|---|---|
| Regression | KRR | Square loss | $\mathbb{R}$ | $(y-u)^2/2$ | $(v)^2/2 + vy$ | $\mathbb{R}$ |
| Regression | $L_p$-Reg. | $L_p$ loss | $\mathbb{R}$ | $\lvert y-u \rvert^p/p$ | $\lvert v \rvert^q/q + vy$ | $\mathbb{R}$ |
| Regression | $L_1$-Reg. | Absolute loss | $\mathbb{R}$ | $\lvert y-u \rvert$ | $vy$ | $-1 \le v \le 1$ |
| Regression | Huber Reg. | Huber loss | $\mathbb{R}$ | $\left((\cdot)^2/2 \square \delta \lvert y - \cdot \rvert \right)(u)$ | $(v)^2/2 + vy$ | $-\delta \le v \le \delta$ |
| Regression | SVR | $\varepsilon$-insensitive | $\mathbb{R}$ | $\left(\lvert \cdot \rvert \square \pi_{\lvert y - \cdot \rvert \le \varepsilon}\right)(u)$ | $\varepsilon \lvert v \rvert + vy$ | $-1 \le v \le 1$ |
| Classification | $L_1$-SVC | Hinge | $\{-1, 1\}$ | $\max\{0, 1-yu\}$ | $vy$ | $-1 \le vy \le 0$ |
| Classification | $L_2$-SVC | Squared hinge | $\{-1, 1\}$ | $\max\{0, 1-yu\}^2/2$ | $(v)^2/2 + vy$ | $vy \le 0$ |
| Classification | KLR | Logistic | $\{-1, 1\}$ | $\log(1 + \exp(-yu))$ | $\mathsf{bEnt}(-vy)$ | $-1 \le vy \le 0$ |

time consumption. Based on the Newton method, (Marteau-Ferey et al., 2019; Meanti et al., 2020) implements Log-Falkon, a fast large-scale KLR, yet has the same memory issue with Falkon. In this paper, we delve into a joint optimization scheme for lightweight kernel models, reaching a remarkable balance between performance, memory usage, and training time. We present an intuitive comparison between the prevalent kernel methods and Joker in Table 1.

### 1.3.2. Dual Coordinate descent algorithms

Coordinate descent methods (CD) iteratively select one variable for optimization while keeping all other variables fixed, aiming to decrease the objective function incrementally. In machine learning, CD has succeeded in the fast training of linear SVM (Hsieh et al., 2008; Dai & Qiu, 2023). Moreover, Shalev-Shwartz & Zhang (2013) proposed a dual optimization framework with coordinate ascent methods for linear models. A Newton-based dual CD method is investigated for unconstrained optimization (Qu et al., 2016). In Joker, we design a unified optimization scheme also using duality. Nevertheless, our work has crucial differences from theirs in two aspects. (i) We focus on kernel models, which are usually ill-conditioned and more challenging in optimization than the linear models. (ii) We employ block coordinate descent (BCD) for more efficient optimization. Note that Tu et al. (2016) and Rathore et al. (2024) also proposed to address KRR with the BCD algorithm. Notably, (Rathore et al., 2024) introduces the Nesterov acceleration technique for BCD and proves its convergence under proper conditions. Beyond their scope, we investigate a general class of kernel models, including but not limited to KRR, KLR, and SVM, and tackle the intricate constrained optimization.

## 2. Joker

Joker focus on convex problem (1) reformulated as:

$$\min_{\boldsymbol{\theta}} \frac{1}{2}\lVert \boldsymbol{\theta} \rVert^2 + \frac{1}{\lambda} \sum_{i=1}^{n} \ell(y_i, \langle \boldsymbol{\theta}, \boldsymbol{\varphi}(\boldsymbol{x}_i) \rangle). \quad (2)$$

We begin with its dual problem (Section 2.1) and present an optimization roadmap (Section 2.2). Inexact Joker is then proposed in Section 2.3.

### 2.1. Joint Optimization Problem by Duality

We first present a direct result for the dual problem of (2).

**Theorem 2.1.** *Let $\xi_y(\cdot) : \mathbb{R} \mapsto \mathbb{R}_+$ defined as $\xi_y(u) := \ell(y, u)$. Then the optimal solution of (2) is given by*

$$\boldsymbol{\theta}^\star = \sum_{i=1}^{n} \alpha_i^\star \boldsymbol{\varphi}(\boldsymbol{x}_i), \quad (3)$$

$$\boldsymbol{\alpha}^\star = \arg\min_{\boldsymbol{\alpha} \in \Omega} \frac{1}{2} \boldsymbol{\alpha}^\top \boldsymbol{K} \boldsymbol{\alpha} + \frac{1}{\lambda} \sum_{i=1}^{n} \xi_{y_i}^* \left(-\lambda \alpha_i\right), \quad (4)$$

*where $\Omega = \{\boldsymbol{\alpha} : -\lambda \alpha_i \in \mathsf{dom}\, \xi_{y_i}^*, i \in [n]\}$ is the feasible region, $\xi_y^*(\cdot)$ is the Fenchel conjugate of $\xi_y(\cdot)$, and $\boldsymbol{K}$ is kernel matrix with $K_{ij} = \langle \boldsymbol{\varphi}(\boldsymbol{x}_i), \boldsymbol{\varphi}(\boldsymbol{x}_j) \rangle$.*

The proof is shown in Appendix A. We first present some important properties. (i) Problem (4) is convex due to the convexity of $\xi_{y_i}^*(\cdot)$. (ii) The strong duality holds according to Slater's condition (Boyd et al., 2004), meaning that if $\boldsymbol{\alpha}^\star$ is optimal for (4), then $\boldsymbol{\theta}^\star$ is optimal for (2). (iii) Based on the closeness and the convexity of the conjugate function (Beck, 2017, Theorem 4.3), the domain of $\xi_{y_i}^*(\cdot)$ should be a closed interval. Therefore, the constraints in (4) are simply box constraints, i.e., $\alpha_i \in [\tau_i^L, \tau_i^U]$ with $-\infty \le \tau_i^L < \tau_i^U \le \infty$. (iv) Problem (4) can be better conditioned than the primal form (2), as the dual Hessian is linearly dependent on $\boldsymbol{K}$ and the primal Hessian is a quadratic form of $\boldsymbol{K}$. Since $\boldsymbol{K}$ is usually ill-conditioned, dual optimization should converge faster than the primal one. This benefit is also noted by Tu et al. (2016) and Rathore et al. (2024).

Theorem 2.1 covers a wide range of kernel models using different loss functions. We list representative ones in Table 2. Notably, problem (4) can easily adapt to some sophisticated loss functions, whereas its primal problem (2) may be tricky. To state this, we introduce the following proposition:

**Algorithm 1:** Trust region (twice-differentialable $f$)

**Input** : Initial point $\boldsymbol{\alpha}_{\mathcal{B}}$, block kernel matrix $\boldsymbol{K}_{\mathcal{B},\mathcal{B}}$, kernel gradient $\bar{\boldsymbol{g}} = \boldsymbol{K}_{\mathcal{B},:}\boldsymbol{\alpha}$, function $f(\cdot)$. Max region size $\Delta_{\max}$, threshold $\eta \in [0, 1/4]$, tolerance $\epsilon > 0$, and max iteration $T_{\mathsf{TR}}$.

$\boldsymbol{\alpha}_{\mathcal{B},0} \leftarrow \boldsymbol{\alpha}_{\mathcal{B}}, \ \Delta_0 \leftarrow \Delta_{\max}/4$;
$\boldsymbol{\tau}_{\mathcal{B}}^U \leftarrow f(\boldsymbol{\alpha}_{\mathcal{B}}).\texttt{upper}, \quad \boldsymbol{\tau}_{\mathcal{B}}^L \leftarrow f(\boldsymbol{\alpha}_{\mathcal{B}}).\texttt{lower}$;
**for** $k = 1$ **to** $T_{\mathsf{TR}}$ **do**

$\quad$ $\boldsymbol{g}_k \leftarrow \bar{\boldsymbol{g}} + \nabla f(\boldsymbol{\alpha}_{\mathcal{B},k}), \boldsymbol{Q}_k \leftarrow \boldsymbol{K}_{\mathcal{B},\mathcal{B}} + \nabla^2 f(\boldsymbol{\alpha}_{\mathcal{B},k})$;
$\quad$ $\boldsymbol{s}_k \leftarrow \texttt{TCG-Steihaug}(\boldsymbol{Q}_k, \boldsymbol{g}_k, \boldsymbol{\alpha}_{\mathcal{B},k}, \boldsymbol{\tau}_{\mathcal{B}}^U, \boldsymbol{\tau}_{\mathcal{B}}^L)$,
$\quad$ i.e. Algorithm 2;
$\quad$ $\rho_k \leftarrow (J(\boldsymbol{\alpha}_{\mathcal{B},k}) - J(\boldsymbol{\alpha}_{\mathcal{B},k} + \boldsymbol{s}_k))/(\mu(\mathbf{0}) - \mu(\boldsymbol{s}_k))$;
$\quad$ **if** $\rho_k < 0.5$ **then**
$\quad\quad$ $\Delta_{k+1} \leftarrow \Delta_k/4$;
$\quad$ **else if** $\rho_k > 0.75$ *and* $\Delta_k - \|\boldsymbol{s}\| < \epsilon$ **then**
$\quad\quad$ $\Delta_{k+1} \leftarrow \min\{2\Delta_k, \Delta_{\max}\}$;
$\quad$ **else**
$\quad\quad$ $\Delta_{k+1} \leftarrow \Delta_k$;
$\quad$ **end**
$\quad$ **if** $\rho_k > \eta$ **then**
$\quad\quad$ $\boldsymbol{\alpha}_{\mathcal{B}} \leftarrow \boldsymbol{\alpha}_{\mathcal{B}} + \boldsymbol{s}$;
$\quad\quad$ $\bar{\boldsymbol{g}} \leftarrow \bar{\boldsymbol{g}} + \boldsymbol{K}_{\mathcal{B},\mathcal{B}}\boldsymbol{s}$;
$\quad$ **end**

**end**

**Proposition 2.2.** *If $\xi_y(\cdot)$ can be written as $\xi_y(u) : z = (\xi_{y,1}\square\xi_{y,2}\square\cdots\square\xi_{y,s})(u)$, then the dual problem of (2) is*

$$\min_{\boldsymbol{\alpha}\in\Omega} \frac{1}{2}\boldsymbol{\alpha}^\top \boldsymbol{K}\boldsymbol{\alpha} + \frac{1}{\lambda}\sum_{i=1}^{n}\sum_{r=1}^{s}\xi^*_{y_i,r}(-\lambda\alpha_i). \qquad (5)$$

*with $\Omega = \{\boldsymbol{\alpha} : -\lambda\alpha_i \in \bigcap_r \mathrm{dom}\,\xi^*_{y_i,r}, (i,r) \in [n]\times[s]\}$.*

*Proof.* The result is simply by induction and the property of infimal convolution: $(\xi_1\square\xi_2)^* = \xi_1^* + \xi_2^*$ (Beck, 2017, Theorem 4.16). $\qquad\square$

For example, Huber loss is used for robust regression, which is defined as

$$\ell(y, u) = \begin{cases} \frac{1}{2}(u-y)^2, & \text{if } |u-y| \le \delta, \\ \delta|u-y| - \frac{1}{2}\delta^2, & \text{Otherwise.} \end{cases}$$

It may be uneasy to optimize its primal problem directly. However, as it can be written as the infimal convolution form: $\xi_y(u) = \left(\frac{1}{2}(\cdot)^2\square\delta|y-\cdot|\right)(u)$, the Fenchel conjugate $\xi_y^*(u)$ is easily derived via Proposition 2.2. Interestingly, Huber loss and square loss (KRR) have almost the same dual problem, with the only difference in the feasible region. The commonalities of the two models imply that they can be solved similarly.

## 2.2. Dual Block Coordinate Descent with Trust Region

Coordinate descent (CD) is particularly suitable for the optimization of (4) for its separable structure (Nutini et al., 2022). The key idea of CD is to optimize one variable while fixing others in each iteration, exhibiting inexpensive computation and storage. Despite its simplicity, it suffers from slow convergence for large-scale kernel machines since it only updates one variable at a time. Thus, we propose updating multiple variables (i.e., a block) simultaneously to leverage the merit of parallel computing, leading to our core solver, Dual Block Coordinate Descent with Trust Region (DBCD-TR). We show the main idea and leave some details in Appendix B.

In each iteration of block coordinate descent, we pick an index set $\mathcal{B} = \{i_1, \cdots, i_{|\mathcal{B}|}\} \subset [n]$ and $\mathcal{B}_{\complement} := [n]\backslash\mathcal{B}$ is the index set associated with the fixed variables. We define $f(\boldsymbol{\alpha}_{\mathcal{B}}) := \sum_{i\in\mathcal{B}}\xi^*_{y_i}(-\lambda\alpha_i)/\lambda$ for simplicity, where $\boldsymbol{\alpha}_{\mathcal{B}}$ is the subvector of $\boldsymbol{\alpha}$ indexed by $\mathcal{B}$. Minimizing (4) w.r.t $\boldsymbol{\alpha}_{\mathcal{B}}$ while fixing $\boldsymbol{\alpha}_{\mathcal{B}_{\complement}}$, we have:

$$\min_{\boldsymbol{\alpha}_{\mathcal{B}}} J(\boldsymbol{\alpha}_{\mathcal{B}}) := \frac{1}{2}\boldsymbol{\alpha}_{\mathcal{B}}^\top \boldsymbol{K}_{\mathcal{B},\mathcal{B}}\boldsymbol{\alpha}_{\mathcal{B}} + \boldsymbol{\alpha}_{\mathcal{B}_{\complement}}^\top \boldsymbol{K}_{\mathcal{B}_{\complement},\mathcal{B}}\boldsymbol{\alpha}_{\mathcal{B}} + f(\boldsymbol{\alpha}_{\mathcal{B}}),$$
$$\text{s.t. } \boldsymbol{\tau}_{\mathcal{B}}^L \le \boldsymbol{\alpha}_{\mathcal{B}} \le \boldsymbol{\tau}_{\mathcal{B}}^U, \qquad (6)$$

There are two challenges in solving subproblem (6): (i) The smoothness of $f(\boldsymbol{\alpha}_{\mathcal{B}})$ is unknown; (ii) even with smooth $f(\boldsymbol{\alpha}_{\mathcal{B}})$, the box constraints are still tricky. For (ii), the projected Newton method (Gafni & Bertsekas, 1984) is a good choice to cope with the box constraints. However, its difficulty is the step size tuning. An improper step size could degrade the projected Newton step (Schmidt et al., 2011; Nutini et al., 2022).

To solve (6), we employ the trust region method (Sorensen, 1982), which is an iterative algorithm for efficient optimization. Denote $\boldsymbol{\alpha}_{\mathcal{B},k}$ the $k$-th iterate of the trust region procedure. The next iterate is given by $\boldsymbol{\alpha}_{\mathcal{B},k+1} := \boldsymbol{\alpha}_{\mathcal{B},k} + \boldsymbol{s}_k$ with a proper step $\boldsymbol{s}_k$ restricted in the trust region $\{\boldsymbol{s} : \|\boldsymbol{s}\| \le \Delta_k\}$, where $\Delta_k$ is the radius. Define a quadratic model function:

$$\mu_k(\boldsymbol{s}) := J(\boldsymbol{\alpha}_{\mathcal{B},k}) + \boldsymbol{g}_k^\top\boldsymbol{s} + \frac{1}{2}\boldsymbol{s}^\top \boldsymbol{Q}_k\boldsymbol{s}. \qquad (7)$$

such that $\mu_k(\boldsymbol{s}) \approx J(\boldsymbol{\alpha}_{\mathcal{B},k} + \boldsymbol{s})$ for $\|\boldsymbol{s}\| \le \Delta_k$. Then we find the step $\boldsymbol{s}_k$ by minimizing an easier quadratic function $\mu_k(\boldsymbol{s})$. The trust region method can overcome two challenges in our problem (6) naturally. Firstly, it is compatible with non-smooth optimization (Baraldi & Kouri, 2025), and secondly, it implicitly tunes the step size by adjusting the radius $\Delta_k$ in each iteration, giving a crucial safeguard for convergence (Baraldi & Kouri, 2024).

We first study the most common scenario where $f$ is twice differentiable, which is applicable to most losses listed in Table 2. The procedure described below is summarized in

**Algorithm 2:** Truncated CG-Steihaug

**Input** : Quadratic model $\boldsymbol{Q}, \boldsymbol{g}$, initial guess $\boldsymbol{\alpha}_{\mathcal{B}}$, region size $\Delta$, bounds $\boldsymbol{\tau}_{\mathcal{B}}^U, \boldsymbol{\tau}_{\mathcal{B}}^L$, tolerance $\varepsilon$.

**Output**: Truncated CG step $\boldsymbol{s}$.

$\boldsymbol{s} = \boldsymbol{0}, \boldsymbol{r} \leftarrow -\boldsymbol{g}, \boldsymbol{d} \leftarrow \boldsymbol{r}, \texttt{r2old} \leftarrow \boldsymbol{r}^\top \boldsymbol{r}$;

**while** *not converged* **do**

  $\omega \leftarrow (\texttt{r2old})/(\boldsymbol{d}^\top \boldsymbol{Q} \boldsymbol{d}), \boldsymbol{s}_{\text{next}} \leftarrow \boldsymbol{s} + \omega \boldsymbol{d}$;

  **if** $\|\boldsymbol{s}_{\text{next}}\| > \Delta$ **then**

    Determines $\omega' > 0$ such that $\|\boldsymbol{s} + \omega' \boldsymbol{d}\| = \Delta$;

    $\boldsymbol{s} \leftarrow \boldsymbol{s} + \omega' \boldsymbol{d}$ and break;

  **else if** $\boldsymbol{s}_{\text{next}}$ *violates box constraints* **then**

    break;

  **end**

  $\boldsymbol{s} \leftarrow \boldsymbol{s}_{\text{next}}, \quad \boldsymbol{r} \leftarrow \boldsymbol{r} - \omega \boldsymbol{Q} \boldsymbol{d}, \quad \texttt{r2new} \leftarrow \boldsymbol{r}^\top \boldsymbol{r}$;

  **if** $\texttt{r2new} \leq \varepsilon$ **then**

    break;

  **end**

  $\nu \leftarrow \texttt{r2new}/\texttt{r2old}, \boldsymbol{d} \leftarrow \boldsymbol{r} + \nu \boldsymbol{d}$;

**end**

**return** $\max\{\min\{\boldsymbol{s}, \boldsymbol{\tau}_{\mathcal{B}}^U - \boldsymbol{\alpha}_{\mathcal{B}}\}, \boldsymbol{\tau}_{\mathcal{B}}^L - \boldsymbol{\alpha}_{\mathcal{B}}\}$;

---

Algorithm 1. In this case, we can construct the quadratic model $\mu_k(\cdot)$ using Taylor expansion. That is, let $\boldsymbol{Q}_k = \boldsymbol{K}_{\mathcal{B},\mathcal{B}} + \nabla^2 f(\boldsymbol{\alpha}_{\mathcal{B},k})$ and $\boldsymbol{g}_k = \boldsymbol{K}_{\mathcal{B},:}\boldsymbol{\alpha} + \nabla f(\boldsymbol{\alpha}_{\mathcal{B},k})$ in (7). Then the "next step" $\boldsymbol{s}_k$ is given by:

$$\boldsymbol{s}_k = \operatorname*{argmin}_{\boldsymbol{\tau}_{\mathcal{B}}^L \leq \boldsymbol{\alpha}_{\mathcal{B},k} + \boldsymbol{s} \leq \boldsymbol{\tau}_{\mathcal{B}}^U} \frac{1}{2} \boldsymbol{s}^\top \boldsymbol{Q}_k \boldsymbol{s} + \boldsymbol{g}_k^\top \boldsymbol{s}, \text{ s.t. } \|\boldsymbol{s}\| \leq \Delta, \tag{8}$$

However, $\boldsymbol{s}_k$ may not be the good enough step when $\mu_k(\boldsymbol{s})$ does not approximate $J(\boldsymbol{\alpha}_{\mathcal{B},k} + \boldsymbol{s})$ well. Considering this, we should evaluate the quality of $\boldsymbol{s}_k$ and only apply $\boldsymbol{\alpha}_{\mathcal{B},k+1} = \boldsymbol{\alpha}_{\mathcal{B},k} + \boldsymbol{s}_k$ for the qualified step, and keep unmoved otherwise, i.e., $\boldsymbol{\alpha}_{\mathcal{B},k+1} = \boldsymbol{\alpha}_{\mathcal{B},k}$. A generic trust region procedure evaluates $\boldsymbol{s}_k$ by the ratio:

$$\rho_k := \frac{J(\boldsymbol{\alpha}_{\mathcal{B},k}) - J(\boldsymbol{\alpha}_{\mathcal{B},k} + \boldsymbol{s}_k)}{\mu_k(\boldsymbol{0}) - \mu_k(\boldsymbol{s}_k)}. \tag{9}$$

A large $\rho_k$ suggests that the objective $J(\cdot)$ is decreased sufficiently, and we tend to accept $\boldsymbol{s}_k$. Specifically, $\boldsymbol{s}_k$ is qualified if $\rho_k > \eta$, where $\eta \in (0, 1/4]$ is the acceptance threshold. On the other hand, we can know that $\mu_k(\boldsymbol{s})$ cannot approximate $J(\boldsymbol{\alpha}_{\mathcal{B},k} + \boldsymbol{s})$ when $\rho_k$ is small. In this case, the radius of the trust region should be reduced, e.g., $\Delta_{k+1} := \Delta_k/4$. Oppositely, we can enlarge the radius in the next iteration to allow a larger step when $\rho_k$ is large.

The subsequent issue is to find an effective solver for (8). The vanilla trust region problem can be solved efficiently with the conjugate gradient method (CG) proposed by Steihaug (1983). However, for (8), extra consideration should be taken on the box constraints. To this end, we propose a heuristic truncated CG-Steihaug method, as shown in Algorithm 2. The key is to terminate the CG procedure if $\boldsymbol{s}$ violates the box constraints or goes beyond the trust region boundary, and finally project $\boldsymbol{s}$ back to the feasible region. Compared with the projected Newton method suggested in (Gafni & Bertsekas, 1984; Nutini et al., 2022), Algorithm 2 computes a truncated CG step instead of the exact inverse $\boldsymbol{Q}^{-1}\boldsymbol{g}$, and thus is more efficient than their projected Newton step. Moreover, Algorithm 2 elegantly embeds the step size tuning into the projected Newton by the nature of the trust region, which also eases the implementation.

Now we consider the complexity of DBCD-TR per iteration. Its space complexity is only $O(|\mathcal{B}|^2)$ lying in the storage of $\boldsymbol{K}_{\mathcal{B},\mathcal{B}}$. In each iteration, Algorithm 2 is repeated $T_{\text{TR}}$ times, resulting in a time complexity of $O(T_{\text{TR}} T_{\text{CG}} |\mathcal{B}|^2)$, where $T_{\text{CG}}$ is the number of CG iterations. In our implementation, $T_{\text{TR}} \leq 50$ and $T_{\text{CG}}$ is generally tiny ($\leq 10$) due to CG's fast convergence and the truncations. The most expensive computation lies in $\boldsymbol{K}_{\mathcal{B},:}\boldsymbol{\alpha}$ with a time complexity of $O(nd|\mathcal{B}|)$ supposing a single kernel evaluation costs $O(d)$ time. This brings us to the next major issue to be overcome.

### 2.3. Inexact **Joker** via Randomized Features

A blueprint to solve the pivotal problem (4) is presented in Section 2.2. If $\boldsymbol{K}_{\mathcal{B},\mathcal{B}}$ and $\boldsymbol{K}_{\mathcal{B},:}\boldsymbol{\alpha}$ are computed exactly in DBCD-TR, we obtain **exact Joker**. However, its bottleneck occurs in computing $\boldsymbol{K}_{\mathcal{B},:}\boldsymbol{\alpha}$ for the $O(nd|\mathcal{B}|)$ time complexity, which becomes a heavy computational burden when $n \geq 10^6$. To alleviate it, we propose **inexact Joker**. The goal is to approximate the exact kernel evaluations with a finite-dimensional mapping $\boldsymbol{\psi}(\cdot) : \mathcal{X} \mapsto \mathbb{R}^M$ with $M \ll n$ such that $\mathcal{K}(\boldsymbol{x}, \boldsymbol{x}') \approx \boldsymbol{\psi}(\boldsymbol{x})^\top \boldsymbol{\psi}(\boldsymbol{x}')$. The following discussion and our implementation are based on the Random Fourier feature (RFF) (Rahimi & Recht, 2007), a well-studied kernel approximation approach. Using Bochner's theorem, one can write a shift-invariant kernel (i.e, the value of $\mathcal{K}(\boldsymbol{x}, \boldsymbol{x}')$ only depends on $\boldsymbol{x} - \boldsymbol{x}'$) as

$$\mathcal{K}(\boldsymbol{x}, \boldsymbol{x}') = \int e^{j \boldsymbol{w}^\top (\boldsymbol{x} - \boldsymbol{x}')} \mathrm{d} p_{\mathcal{K}}(\boldsymbol{w}) = \mathbb{E}_{\boldsymbol{w}}[\zeta_{\boldsymbol{w}}(\boldsymbol{x}) \bar{\zeta}_{\boldsymbol{w}}(\boldsymbol{x}')],$$

where $j = \sqrt{-1}$ denotes imaginary unit, $p_{\mathcal{K}}$ is a proper probability distribution associated with the kernel $\mathcal{K}$, and $\zeta_{\boldsymbol{w}}(\boldsymbol{x}) = \exp(-j\boldsymbol{w}^\top \boldsymbol{x})$. Considering $\mathcal{K}$ is real-valued, we can further derive

$$\mathcal{K}(\boldsymbol{x}, \boldsymbol{x}') = \mathbb{E}_{\boldsymbol{w} \sim p_{\mathcal{K}}, b \sim U_{[0,2\pi]}} [2 \cos(\boldsymbol{w}^\top \boldsymbol{x} + b) \cos(\boldsymbol{w}^\top \boldsymbol{x}' + b)],$$

where $U_{[0,2\pi]}$ denotes the uniform distribution on $[0, 2\pi]$. Based on the Monte Carlo method, $\boldsymbol{\psi}(\boldsymbol{x})$ can be defined as:

$$\boldsymbol{\psi}(\boldsymbol{x}) = \sqrt{\frac{2}{M}} \cos(\boldsymbol{W}\boldsymbol{x} + \boldsymbol{b}), b_i \sim U_{[0,2\pi]}, \boldsymbol{w}_i \sim p_{\mathcal{K}}, \tag{10}$$

**Algorithm 3:** DBCD-TR for problem (4)

**Input** : Kernel $\mathcal{K}$, feasible region $\Omega$, function $\xi_y(\cdot)$, parameter $\lambda$, block size $b$, max iteration $T$.

**Output :** The multiplier $\boldsymbol{\alpha}$ (predictor $\boldsymbol{\theta}$)

Initialize $\boldsymbol{\alpha} \in \Omega$, partition $[n]$ into blocks $\mathcal{B}_1, \cdots, \mathcal{B}_m$;

**if** *using inexact model* **then**

 Sample $\boldsymbol{W} \in \mathbb{R}^{M \times d} \sim p_{\mathcal{K}}, \boldsymbol{b} \in \mathbb{R}^M \sim U_{[0,2\pi]}$;

 Define map $\boldsymbol{\psi}(\boldsymbol{x}) := \cos(\boldsymbol{W}\boldsymbol{x} + \boldsymbol{b})$;

 Initialize $\boldsymbol{\theta}$ such that $\boldsymbol{\theta} = \sum_{i=1}^n \alpha_i \boldsymbol{\psi}(\boldsymbol{x}_i)$.

**end**

**for** $t = 1, 2, \cdots, T$ **do**

 Randomly pick a block $\mathcal{B} \in \{\mathcal{B}_1, \cdots, \mathcal{B}_m\}$;

 Let $f(\boldsymbol{\alpha}_{\mathcal{B}}) := \sum_{i \in \mathcal{B}} \xi^*_{y_i}(-\lambda\alpha_i)/\lambda$;

 **if** *using inexact model* **then**

  $\boldsymbol{K}_{\mathcal{B},\mathcal{B}} \leftarrow \boldsymbol{\psi}(\boldsymbol{X}_{\mathcal{B}})^\top \boldsymbol{\psi}(\boldsymbol{X}_{\mathcal{B}})$;

  $\boldsymbol{K}_{\mathcal{B},:}\boldsymbol{\alpha} \leftarrow \boldsymbol{\psi}(\boldsymbol{X}_{\mathcal{B}})^\top \boldsymbol{\theta}$;

 **end**

 $\boldsymbol{\alpha}_{\mathcal{B}}^{\text{new}} \leftarrow \texttt{TrustRegion}(\boldsymbol{\alpha}_{\mathcal{B}}, \boldsymbol{K}_{\mathcal{B},\mathcal{B}}, \boldsymbol{K}_{\mathcal{B},:}\boldsymbol{\alpha}, f)$ to solve problem (6), i.e., Algorithm 1;

 **if** *using inexact model* **then**

  $\boldsymbol{\theta} \leftarrow \boldsymbol{\theta} + \boldsymbol{\psi}(\boldsymbol{X}_{\mathcal{B}})(\boldsymbol{\alpha}_{\mathcal{B}}^{\text{new}} - \boldsymbol{\alpha}_{\mathcal{B}})$;

 **end**

 $\boldsymbol{\alpha}_{\mathcal{B}} \leftarrow \boldsymbol{\alpha}_{\mathcal{B}}^{\text{new}}$;

**end**

**return** $\boldsymbol{\alpha}$ (and $\boldsymbol{\theta}$ if using inexact model);

---

where $\cos(\cdot)$ is applied element-wise, $\boldsymbol{W} \in \mathbb{R}^{M \times d}, \boldsymbol{b} \in \mathbb{R}^M$ and each row $\boldsymbol{w}_i, b_i$ are sampled from the probability distributions $p_{\mathcal{K}}(\boldsymbol{w})$ and $U_{[0,2\pi]}$, respectively. Gaussian kernel $\mathcal{K}(\boldsymbol{x}, \boldsymbol{x}') = \exp(-\|\boldsymbol{x} - \boldsymbol{x}'\|^2/(2\sigma^2))$ is a widely used kernel in RFF, which corresponds to $p_{\mathcal{K}}(\boldsymbol{w}) = \mathcal{N}(\mathbf{0}, \sigma^2 \boldsymbol{I})$, i.e., the Gaussian with zero mean and covariance $\sigma^2 \boldsymbol{I}$. Note that although RFF is initially proposed for shift-invariant kernels, it has been sufficiently developed to diverse kernels, such as dot-product kernels and additive kernels. One can find a comprehensive summary of RFF for various kernels in the latest report of (Dai et al., 2014).

Now we use a new kernel $\mathcal{K}_{\text{rff}}(\boldsymbol{x}, \boldsymbol{x}') := \boldsymbol{\psi}(\boldsymbol{x})^\top \boldsymbol{\psi}(\boldsymbol{x}')$ to replace the exact one $\mathcal{K}$. Then the kernel matrix becomes $K_{ij} = \boldsymbol{\psi}(\boldsymbol{x}_i)^\top \boldsymbol{\psi}(\boldsymbol{x}_j)$. Let $\boldsymbol{\varphi}(\boldsymbol{x}) := \boldsymbol{\psi}(\boldsymbol{x})$ in Theorem 2.1, we obtain $\boldsymbol{\theta} = \sum_{i=1}^n \alpha_i \boldsymbol{\psi}(\boldsymbol{x}_i)$, leading to

$$\boldsymbol{K}_{\mathcal{B},:}\boldsymbol{\alpha} = \sum_{i=1}^n \boldsymbol{\psi}(\boldsymbol{X}_{\mathcal{B}})^\top \boldsymbol{\psi}(\boldsymbol{x}_i)\alpha_i = \boldsymbol{\psi}(\boldsymbol{X}_{\mathcal{B}})^\top \boldsymbol{\theta}. \quad (11)$$

where $\boldsymbol{\psi}(\boldsymbol{X}_{\mathcal{B}}) := [\boldsymbol{\psi}(\boldsymbol{x}_i)]_{i \in \mathcal{B}}$. Therefore, time complexity of evaluating $\boldsymbol{K}_{\mathcal{B},:}\boldsymbol{\alpha}$ is reduced to $O(Md|\mathcal{B}|)$. To implement this, we must maintain the weight vector $\boldsymbol{\theta}$ during the optimization. Once $\boldsymbol{\alpha}_{\mathcal{B}}$ is updated, $\boldsymbol{\theta}$ is updated with only $O(M|\mathcal{B}|)$ time complexity:

$$\boldsymbol{\theta}^{\text{new}} := \boldsymbol{\theta}^{\text{old}} + \sum_{i \in \mathcal{B}} (\alpha_i^{\text{new}} - \alpha_i^{\text{old}})\boldsymbol{\psi}(\boldsymbol{x}_i), \quad (12)$$

*Table 3.* Complexity comparison, $|\mathcal{B}| \ll M \ll n$. (Log)Falkon has extra setup time of $O(M^3)$ and post-process time of $O(M^2)$. $\mathcal{M}_{\text{ep2}}$ is the memory cost of EigenPro2 (Ma & Belkin, 2019).

| Methods | Space | Operations per epoch |
|---|---|---|
| (Log)Falkon | $M^2 + Md$ | $nMd$ |
| EigenPro3 | $\mathcal{M}_{\text{ep2}} + Md$ | $nMd + \frac{n}{|\mathcal{B}|}O(M^2)$ |
| Exact Joker | $|\mathcal{B}|^2$ | $n^2d + \frac{n}{|\mathcal{B}|}(d|\mathcal{B}|^2)$ |
| Inexact Joker | $|\mathcal{B}|^2 + Md$ | $nMd + \frac{n}{|\mathcal{B}|}(M|\mathcal{B}|^2 + Md|\mathcal{B}|)$ |

Inexact Joker needs extra memory of $O(Md)$ to store $\boldsymbol{W}$. Notably, increasing $M$ generally produces better approximation and performance, but also costs more time and storage. One can find a theoretical guide for setting $M$ in (Lanthaler & Nelsen, 2023). Up to this point, we can present the complete DBCD-TR procedure in Algorithm 3.

Finally, we justify why we do not consider the Nyström method in the proposed Joker, although it is the preferred approximation method in many works such as (Yang et al., 2012; Rudi et al., 2017). In short, Nyström method is unsuitable in our scenario due to its heavy computation and storage in each iteration. Assume $\boldsymbol{Z} \in \mathcal{X}^M$ are the Nyström centers and $M$ is the number of centers. Then the non-linear map becomes $\boldsymbol{\psi}(\boldsymbol{x}) := \boldsymbol{L}^{-1}\mathcal{K}(\boldsymbol{Z}, \boldsymbol{x})$, where $\boldsymbol{L}$ is the Cholesky factor of $\mathcal{K}(\boldsymbol{Z}, \boldsymbol{Z})$. Therefore, to utilize the Nyström method, one should first compute $\boldsymbol{L}$ with $O(M^3)$ time complexity and store it with $O(M^2)$ space complexity. When updating a block $\mathcal{B}$ during training, the computation of $\boldsymbol{\psi}(\boldsymbol{x})$ costs $O(M^2 + Md)$ time, where $O(M^2)$ is from the inverse of the triangular matrix $\boldsymbol{L}$. In other words, the time complexity per block update is at least $O(M^2 + Md)$, so it is too expensive when $M$ is large. Unfortunately, the existing works (Rudi et al., 2017; Abedsoltan et al., 2023) have shown that a large $M$ is necessary to yield a satisfying performance. Therefore, the Nyström method does not fit Joker. In contrast, RFF is more efficient and scalable. We compare the complexity of different methods in Table 3.

## 3. Practical Instances

This section presents some example models of Joker and the associated implementation issues. Additionally, SVR is also discussed in Appendix B.2.

### 3.1. Simple cases: KRR, Huber regression and $L_2$-SVC

Based on Table 2 and Theorem 2.1, the dual problems of KRR, Huber, and $L_2$-SVC share the same form:

$$\min_{\boldsymbol{\alpha} \in \Omega} \frac{1}{2}\boldsymbol{\alpha}^\top (\boldsymbol{K} + \lambda\boldsymbol{I})\boldsymbol{\alpha} - \boldsymbol{y}^\top \boldsymbol{\alpha}, \quad (13)$$

where $\Omega = \mathbb{R}^n$ for KRR, $\Omega = \{\boldsymbol{\alpha} : \|\boldsymbol{\alpha}\|_\infty \leq \delta/\lambda\}$ for Huber, and $\Omega = \{\boldsymbol{\alpha} : \alpha_i y_i \geq 0\}$ for $L_2$-SVC. That is, the

three models only have differences in the feasible region. Their similarity eases the practical implementation. Due to the simplicity of the quadratic functions, their convergence is usually fast. The empirical results (Table 4) show that the elapsed time of these three models is close, suggesting that they can achieve comparable speed. That is, Joker fills the potential efficiency gap between different kernel models.

### 3.2. A Complicated case: KLR

The case of KLR is more complicated and has some practical problems. From now on, we define $\widehat{\boldsymbol{\alpha}} := \boldsymbol{y} \odot \boldsymbol{\alpha}$, where $\odot$ denotes the element-wise product. We obtain the dual problem for KLR:

$$\min_{\boldsymbol{\alpha} \in \Omega} \frac{1}{2} \boldsymbol{\alpha}^\top \boldsymbol{K} \boldsymbol{\alpha} + \sum_{i=1}^n \widehat{\alpha}_i \log \widehat{\alpha}_i + (\lambda^{-1} - \widehat{\alpha}_i) \log(\lambda^{-1} - \widehat{\alpha}_i).$$

The feasible region is $\Omega = \{\boldsymbol{\alpha} : 0 < \widehat{\alpha}_i < 1/\lambda\}$. Due to $\lim_{x \to 0} x \log x = 0$, it can be extended to $0 \le \widehat{\alpha}_i \le 1/\lambda$ by defining $0 \log 0 = 0$. However, a problem arises when $\widehat{\alpha}_i$ is near the boundary. Consider the gradient and Hessian:

$$\nabla f(\widehat{\boldsymbol{\alpha}}_\mathcal{B})_i = \log(\widehat{\alpha}_i) - \log(\lambda^{-1} - \widehat{\alpha}_i),$$
$$\nabla^2 f(\widehat{\boldsymbol{\alpha}}_\mathcal{B})_{ii} = [\widehat{\alpha}_i (1 - \lambda \widehat{\alpha}_i)]^{-1}.$$

We can see that they are unbounded at 0 and $1/\lambda$. This poses two issues. First, the quadratic model in (8) becomes ill-conditioned near the boundary, causing a rapid shrinkage of the trust region (i.e., $\Delta \to 0$) and slow convergence (Baraldi & Kouri, 2025). The second issue is the potential catastrophic cancellation (Yu et al., 2011). When $\widehat{\alpha}_i \approx 0$, the result of $1/\lambda - \widehat{\alpha}_i$ may be inaccurate because of the limited precision of the computer, further leading to incorrect logarithms in the gradient computation.

To mitigate these issues, we first redefine the feasible region as $\varepsilon \le \widehat{\alpha}_i \le 1/\lambda - \varepsilon$, where $\varepsilon$ is the distance from $1/\lambda$ to its next smaller floating-point number. In this way, $1/\lambda - \widehat{\alpha}_i$ will always be precise for all feasible $\widehat{\alpha}_i$. Moreover, we utilize a modified Hessian $\widetilde{H}_{ii} = \min(\nabla^2 f(\widehat{\boldsymbol{\alpha}}_\mathcal{B})_{ii}, \varepsilon^{-1/2})$ to avoid the ill-conditioned model. This also alleviates possible catastrophic cancellation when computing the frequent operation $\boldsymbol{Q}\boldsymbol{s}$. Finally, we increase the block size as suggested in (Nutini et al., 2022) to overcome the slow convergence issue. We found that these strategies significantly improve the numerical stability and convergence speed.

## 4. Experiments

To highlight that Joker can obtain promising performance under a limited computational budget, we conduct experiments on a machine with a single consumer GPU (NVIDIA RTX 3080, 10GB) and 64GB RAM. The experiments *always use single precision* unless otherwise specified. The

implementation[2] of Joker is based on PyTorch without extra acceleration libraries.

The used datasets cover both regression (MSD, HEPC) and classification tasks (SUSY, HIGGS, CIFAR-5M) with the sample size ranging from $10^5$ to $10^7$. They are frequently used in the literature of large-scale kernel methods, and the details are summarized in Appendix C. The largest dataset, CIFAR-5M, has 10 classes and 5 million samples, each with 3072 dimensions, barely fitting within the machine's RAM. All datasets are normalized using the z-score trick.

Using Joker's framework, we implement three regressors: **KRR**, **Huber**, and **SVR**, and two classifiers: ($L_2$-)**SVC** and **KLR**. The compared methods are their state-of-the-art counterparts, including Falkon (KRR), EigenPro3 (KRR), LogFalkon (KLR), ThunderSVM (SVC, SVR). We utilize one-versus-rest (OVR) in Joker to support multi-class problems. OVR is also available in ThunderSVM, while LogFalkon only supports binary classification (Meanti et al., 2020) and thus is not applicable on CIFAR-5M, a 10-class dataset. The kernels used in the experiments include the Gaussian and Laplacian kernels, which are two prevalent choices in practice. The regularization parameter $\lambda$ in Joker is tuned from $\{2^i : i = -7, -6, \cdots, 7\}$ via grid search. According to (Nutini et al., 2022), increasing the block size reduces the needed iterations for the convergence of BCD. In most cases, we employ the inexact Joker models with the block size $|\mathcal{B}| = 512$ considering the trade-off between total training time and memory consumption. Except for MSD, we can employ the exact Joker with block size $|\mathcal{B}| = 2048$ due to its relatively small sample size. Exact Joker needs not to store RFF random matrix $\boldsymbol{W}$ and can afford a larger block size than the inexact Joker. We also increase the block size to 1024 for Joker-KLR to accelerate convergence. Falkon and LogFalkon can be run with $M = 2.5 \times 10^4$ within the limited memory. Details of further parameter settings are shown in Appendix C.

### 4.1. Results and Analysis

Table 4 shows the result of the performance comparison. Note that ThunderSVM and Joker-SVM both include SVR and SVC cases, where SVR is applied to MSD and HEPC datasets, and SVC is applied to others. We observe that Joker-based methods reach the lowest memory usage on all tested datasets. To obtain equivalent performance in (Meanti et al., 2020) on HIGGS ($\approx 74.22\%$ with $M = 10^5$, unaffordable for the used machine), Joker only needs 1.9GB GPU memory, saving at least 95% storage costs comparing (Meanti et al., 2020). Despite low memory, Joker still outperforms the state-of-the-art methods in most cases, presenting almost no accuracy sacrifice. Joker's time is signifi-

---

[2]Code available at GitHub: https://github.com/Apple-Zhang/Joker-paper.

*Table 4.* The performance comparison on regression (MSD, HEPC) and classification (SUSY, HIGGS, CIFAR-5M) datasets. "↓": lower is better, and vice versa for "↑". "NA": meaning not applicable. "Timeout": the running time exceeds the limit of 1 week. "†": using the double floating-point precision. Data below each term indicates the time and peak GPU memory consumption. ThunderSVM and Joker-SVM apply the SVR model on MSD and HEPC datasets and the SVC model on others. The top-2 results are highlighted in bold.

| Methods | MSD ($n \approx 0.5M$) | HEPC ($n \approx 2M$) | SUSY ($n \approx 5M$) | | HIGGS ($n \approx 11M$) | | CIFAR-5M |
|---|---|---|---|---|---|---|---|
| | rel. error ($\times 10^{-3}$, ↓) | RMSE ($\times 10^{-2}$, ↓) | AUC (%, ↑) | ACC (%, ↑) | AUC (%, ↑) | ACC (%, ↑) | ACC (%, ↑) |
| Falkon | **4.4984**±0.0013 (6min, 9.8GB) | 5.4642±0.0178[†] (19min, 9.9GB) | 87.61±0.00 (9min, 6.0GB) | 80.38±0.00 | 80.90±0.09 (31min, 9.9GB) | 73.42±0.07 | 68.24±0.33 (1.9h, 9.9GB) |
| LogFalkon | NA | NA | 87.77±0.05 (13min, 6.5GB) | 80.49±0.00 | 80.43±0.02 (45min, 9.9GB) | 73.04±0.02 | NA |
| EigenPro3 | 4.5512±0.0047 (1.0h, 1.6GB) | 5.0417±0.0014 (2.2h, 1.8GB) | 86.99±0.01 (2.2h, 1.7GB) | 80.08±0.01 | 79.74±0.13 (18h, 7.0GB) | 72.46±0.06 | 72.94±0.00 (80h, 6.9GB) |
| ThunderSVM | 4.6431±0.0257 (3.2h, 5.0GB) | 6.0834±0.0847 (3.2h, 8.0GB) | 79.32±0.01 (15h, 7.8GB) | 80.22±0.01 | Timeout | | Timeout |
| Joker-KRR | **4.4868**±0.0012 (35min, **0.7GB**) | **4.7170**±0.0007 (31min, **1.5GB**) | 87.63±0.00 (25min, **1.2GB**) | 80.41±0.01 | 81.94±0.17 (1.0h, **1.9GB**) | 74.03±0.14 | 73.32±0.01 (2.1h, **5.3GB**) |
| Joker-Huber | 4.5058±0.0109 (36min, **0.7GB**) | **4.7160**±0.0004 (36min, **1.5GB**) | 87.64±0.00 (23min, **1.2GB**) | 80.41±0.01 | 81.83±0.27 (57min, **1.9GB**) | 74.01±0.21 | 73.66±0.01 (2.1h, **5.3GB**) |
| Joker-SVM | 4.6004±0.0073 (35min, **0.7GB**) | 4.8376±0.0342 (27min, **1.5GB**) | 87.72±0.01 (25min, **1.2GB**) | **80.44**±0.02 | **82.40**±0.06 (56min, **1.9GB**) | **74.41**±0.05 | **74.47**±0.02 (2.0h, **5.3GB**) |
| Joker-KLR | NA | NA | **87.73**±0.01 (1.1h, 1.7GB) | 80.42±0.01 | **82.11**±0.03 (1.6h, 2.6GB) | **74.17**±0.01 | **74.88**±0.08 (3.2h, 5.9GB) |

cantly lower than EigenPro3 and ThunderSVM. Falkon-based methods are the fastest and have a substantial gap compared to EigenPro3 and ThunderSVM. However, Joker alleviates this gap. Specifically, EigenPro3 and Thunder-SVM use at least 10x training time compared to Falkon on MSD, and Joker reduces it to 5x time. On the other hand, EigenPro3 needs 36x time (18 hours) of Falkon (0.5 hour), and Joker reduces such gap to 2x time. Thus, Joker obtains comprehensively better results under the same hardware conditions, and achieves a good trade-off between memory, time, and accuracy, demonstrating that Joker is highly scalable for large-scale learning.

The performance of various models in Joker is worth noting. The exact Joker-KRR obtains the best performance on MSD using less than 1GB of memory, showing the efficacy of the exact Joker on relatively small datasets. Joker-Huber surpasses other regression models on the HEPC dataset, which may benefit from its robustness. Joker-SVM shows outstanding performance on the classification tasks. ThunderSVM is the most time-consuming method. Particularly, it converges slowly when its penalty parameter $c$ (equivalent to $1/\lambda$ in (2)) is large. Such inefficiency may stem from the SMO solver, which updates only two variables at each iteration, significantly slower than DBCD-TR. In comparison, Joker-SVM converges fast when utilizing large penalty $c = 1/\lambda$ and always outperforms ThunderSVM with less time and memory. We use a larger block size, $|\mathcal{B}| = 1024$, in Joker-KLR rather than $|\mathcal{B}| = 512$ in other

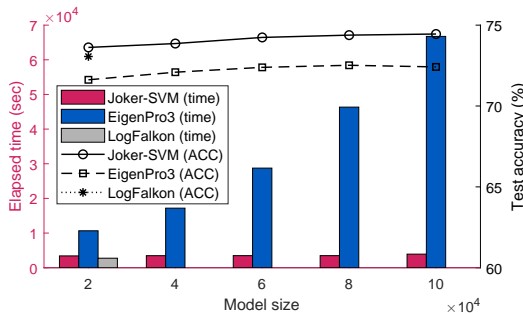

*Figure 1.* Performance versus the model size on HIGGS.

inexact Joker models to overcome the slow convergence caused by the ill-conditioned Hessian. Thus, Joker-KLR takes more time and memory than other Joker models, Despite the ill-conditioning issue, Joker-KLR never meets numerical error degrading the performance in single precision, and still obtains promising accuracy.

Figure 1 illustrates the time and performance versus the model size. In general, a larger model size yields better performance. Joker obtains the best accuracy even with the smallest model. The time of EigenPro3 increases rapidly with the model size, while Joker's time is almost unaffected. In spite of the lowest elapsed time, Falkon fails to scale up the model size due to its expensive memory cost.

A comparison of the training progress of different models is shown in Figure 2. The curves of Joker are not mono-

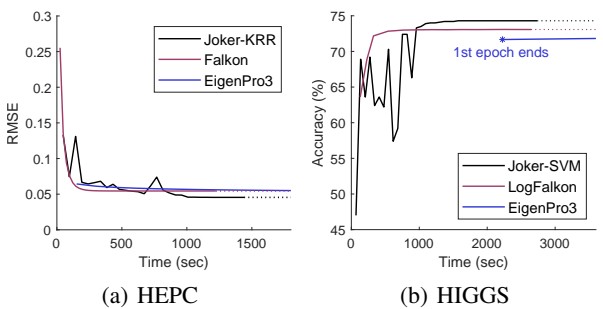

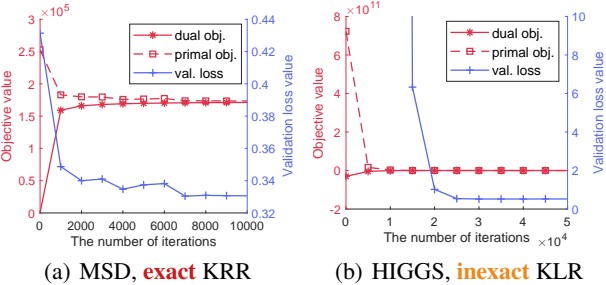

*Figure 2.* Test performance versus time.

*Figure 3.* The primal, dual objectives, and validation loss of Joker versus the iteration steps.

tonical since Joker optimizes the dual problem, which does not guarantee the monotonic decrease of the primal objective. Nonetheless, Joker eventually converges with the best performance. HEPC is a challenging regression dataset, where Falkon obtains suboptimal results using the single precision (RMSE $\approx 17 \times 10^{-2}$). It is mitigated by using double precision (RMSE $\approx 5.4 \times 10^{-2}$). The reason may be the loss of precision caused by matrix decomposition, while Joker and EigenPro3 are free of this issue and obtain better results than Falkon. Nonetheless, the slow convergence of EigenPro3 makes it less competitive than Joker. EigenPro3 just finished the first epoch on HIGGS when Joker and LogFalkon are nearly convergent.

We illustrate the convergence pattern of Joker in Figure 3. The proposed method optimizes (4), i.e., minimizing the negative of the dual objective. Therefore, the dual objective keeps increasing, which fits the pattern shown in the figures. The primal objective is always larger than or equal to the dual objective due to weak duality. As mentioned before, the primal objective and loss may not decrease monotonically. Nonetheless, Figure 3 shows their major trend of decreasing. With sufficient iterations, dual and primal objectives tend to get close and merge, indicating the convergence of Joker.

## 5. Conclusion and Future Directions

Scalability is a crucial issue for kernel methods. In this paper, we propose a novel optimization framework for large-scale kernel methods named Joker, breaking the memory bottleneck and pushing the development of models beyond KRR. The proposed solver, DBCD-TR, provides a modern and efficient solution to dual optimization in kernel machines. We show the effectiveness of Joker on a variety of kernel methods, including KRR, SVM, KLR, etc. Even with consumer hardware and limited memory, Joker obtains state-of-the-art performance within acceptable training time, making the low-cost kernel methods possible in practice.

Regarding future work, generalizations of model (2) can be explored, e.g., multi-class SVM proposed by (Crammer

& Singer, 2001) and softmax regression, whose equality-constrained dual problems are the major issue. In addition, the convergence speed of DBCD-TR is still unclear. Existing theoretical results, e.g., (Richtárik & Takáč, 2014; Nutini et al., 2015; 2022), suggest that DBCD-TR has at least a linear convergence rate. A sharper rate is worth exploring. We feel it is non-trivial and leave it as future work.

## Acknowledgment

This work was supported in part by the National Natural Science Foundation of China under Grant 62476175 and Grant 62272319, and in part by the Natural Science Foundation of Guangdong Province (Grant 2023A1515010677, 2024A1515011637, 2023B1212060076) and Science and Technology Planning Project of Shenzhen Municipality under Grant JCYJ20220818095803007 and JCYJ20240813142206009.

## Impact Statement

This work improves the optimization framework of kernel machines. It significantly reduces the computational requirement and expense of large-scale kernel methods. Comparing the existing methods that may require top-line hardware (e.g., A100, V100 GPU), the proposed method is more friendly for weaker hardware and could be widely applied. Moreover, the proposed techniques support various kernel models beyond KRR, presenting a more comprehensive solution for kernel learning and related applications. To conclude, this work lowers the cost of large-scale learning, widens the application scope of kernel methods, and makes it more accessible to the public.

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

# Appendix

## A. Proof of Theorem 2.1

*Proof.* Note that problem (2) is equivalent to:

$$\min_{\boldsymbol{\theta}, \boldsymbol{u}} \frac{1}{2} \langle \boldsymbol{\theta}, \boldsymbol{\theta} \rangle + \frac{1}{\lambda} \sum_{i=1}^{n} \xi_{y_i}(u_i), \quad \text{s.t. } \langle \boldsymbol{\theta}, \boldsymbol{\varphi}(\boldsymbol{x}_i) \rangle = u_i. \tag{A.1}$$

Introducing the Lagrange multipliers $\boldsymbol{\alpha} \in \mathbb{R}^n$, we can obtain the Lagrangian function:

$$L(\boldsymbol{\theta}, \boldsymbol{u}, \boldsymbol{\alpha}) = \frac{1}{2} \langle \boldsymbol{\theta}, \boldsymbol{\theta} \rangle + \frac{1}{\lambda} \sum_{i=1}^{n} \xi_{y_i}(u_i) + \sum_{i=1}^{n} \alpha_i(\langle \boldsymbol{\theta}, \boldsymbol{\varphi}(\boldsymbol{x}_i) \rangle - u_i). \tag{A.2}$$

The first-order Karush-Kuhn-Tucker (KKT) condition gives

$$\boldsymbol{\theta}^{\star} = \arg\min_{\boldsymbol{\theta}} L(\boldsymbol{\theta}, \boldsymbol{u}, \boldsymbol{\alpha}) = \sum_{i=1}^{n} \alpha_i \boldsymbol{\varphi}(\boldsymbol{x}_i), \tag{A.3}$$

$$\boldsymbol{u}^{\star} = \arg\min_{\boldsymbol{u}} L(\boldsymbol{\theta}, \boldsymbol{u}, \boldsymbol{\alpha}) = \arg\min_{\boldsymbol{u}} \sum_{i=1}^{n} \frac{1}{\lambda} \xi_{y_i}(u_i) + \alpha_i u_i = -\frac{1}{\lambda} \arg\max_{\boldsymbol{u}} \sum_{i=1}^{n} -\lambda \alpha_i u_i - \xi_{y_i}(u_i). \tag{A.4}$$

Recall the definition of the Fenchel conjugate, the maximum in (A.4) is $\sum_{i=1}^{n} \xi_{y_i}^*(-\lambda \alpha_i)$. Therefore, the dual problem is

$$\max_{\boldsymbol{\alpha}} \min_{\boldsymbol{u}, \boldsymbol{\theta}} L(\boldsymbol{\theta}, \boldsymbol{u}, \boldsymbol{\alpha}) = \max_{\boldsymbol{\alpha}} -\frac{1}{2} \sum_{i=1}^{n} \sum_{j=1}^{n} \alpha_i \alpha_k \langle \boldsymbol{\varphi}(\boldsymbol{x}_i), \boldsymbol{\varphi}(\boldsymbol{x}_j) \rangle - \frac{1}{\lambda} \sum_{i=1}^{n} \xi_{y_i}^*(-\lambda \alpha_i), \quad \text{s.t. } -\lambda \alpha_i \in \text{dom } \xi_{y_i}^*. \tag{A.5}$$

Or equivalently, by applying a negative sign to the objective function, we have the minimization problem:

$$\boldsymbol{\alpha}^{\star} = \arg\min_{\boldsymbol{\alpha} \in \Omega} \frac{1}{2} \boldsymbol{\alpha}^{\top} \boldsymbol{K} \boldsymbol{\alpha} + \frac{1}{\lambda} \sum_{i=1}^{n} \xi_{y_i}^*(-\lambda \alpha_i), \quad \text{where } \Omega = \{\boldsymbol{\alpha} \in \mathbb{R}^n : -\lambda \alpha_i \in \text{dom } \xi_{y_i}^*\}. \tag{A.6}$$

Equations (A.3) and (A.6) give the primal-dual relationship, which completes the proof. $\square$

## B. Details of Dual Block Coordinate Descent with Trust Region (DBCD-TR)

### B.1. Strategies of Block Coordinate Descent

As pointed out by Nutini et al. (2022), there are several aspects to consider when designing the BCD algorithm:

- *(i) How to obtain the block candidates?*

- *(ii) How to select the block size?*

- *(iii) How to choose the block to update?*

- *(iv) How to update the block?*

In fact, (iv) has been discussed in the main paper. Here we elaborate on the other three aspects.

**(i):** We adopt the *fixed block* strategy, i.e., the potential selected block is static during the optimization. Specifically, the block candidates are obtained by partitioning the index set $\{1, \cdots, n\}$. Although Nutini et al. (2022) suggests "variable blocks" (the block is dynamically constructed during the optimization) for its faster convergence, its computational expense is higher (e.g., the gradient of all samples should be maintained during optimization). So we choose the fixed block strategy for its simplicity, which also shows satisfactory efficiency in practice.

**(ii):** Increasing block size generally decreases the iterations to reach convergence of BCD (Nutini et al., 2022), but it may increase the computation cost and memory usage of each iteration. So we should find a trade-off between the storage

$S = |\mathcal{B}|^2$ and the time $T_{\text{BCD}} t_{\text{inner}}$, where $T_{\text{BCD}}$ is the number of BCD iterations and $t_{\text{inner}}$ the time complexity of each inner iteration. We have shown that $t_{\text{inner}} = O(|\mathcal{B}|^2)$ in the main paper. For $T_{\text{BCD}}$, Nutini et al. (2022) points out that larger block sizes generally lead to faster convergence. (Richtárik & Takáč, 2014; Nutini et al., 2015) show a linear convergence rate of proximal gradient descent for $\sigma$-strongly convex functions:

$$T_{\text{BCD}} \le O\left(\frac{Ln}{\sigma|\mathcal{B}|} \log\left(\frac{1}{\varepsilon}\right)\right), \tag{A.7}$$

where $L$ is the sum of Lipschitz constants of all blocks. However, this result is not tight. Borrowing the analysis from (Nutini et al., 2022), DBCD-TR adopts the "optimal updates" paradigm, which should be faster than (A.7), even possibly superlinear, but the precise rate is still unclear. Despite this, we still use (A.7) as a tool to analyze the block size setting. In our experiments, we find that $|\mathcal{B}| = 512$ is a balanced choice for KRR, SVC, SVR and Huber. However, because of the ill-conditioned problem, i.e., extremely large $L$ in (A.7), the convergence of Joker-KLR is slower than other Joker models. To overcome it, except for the numerical techniques mentioned in the main paper, increasing the block size is also feasible (by reducing the upper bound of $T_{\text{BCD}}$). Regarding this, we set $|\mathcal{B}| = 1024$ for Joker-KLR in our experiments.

**(iii):** Based on the fixed block strategy, we randomly select the block from the candidates with equal probability. Its advantage over the cylic approach is that it can avoid the potential bias of the fixed selection order. The alternative and probably better strategy is the greedy approach, e.g. Gauss-Southwell rules presented in (Nutini et al., 2022). However, they need to evaluate the gradient (and possibly the Hessian) of all blocks, which is expensive in the large-scale data scenario.

### B.2. Discussion of Nonsmoothness: SVR as an Example

Nonsmoothness is an intrinsic issue for trust region methods. The universal solutions include proximal gradient descent (Aravkin et al., 2022) and proximal Newton projected step (sometimes intractable) (Nutini et al., 2022). $\ell_1$-norm and piecewise linear functions are typical examples of nonsmooth functions. The dual problem of SVR is a representative one of the former:

$$\min_{\boldsymbol{\alpha}} \frac{1}{2}\boldsymbol{\alpha}^\top \boldsymbol{K}\boldsymbol{\alpha} - \boldsymbol{y}^\top \boldsymbol{\alpha} + \varepsilon\|\boldsymbol{\alpha}\|_1, \quad \text{s.t. } \|\boldsymbol{\alpha}\|_\infty \le \frac{1}{\lambda}. \tag{A.8}$$

A commonly used trick was used in (Chang & Lin, 2011) and (Schmidt, 2010). That is, take $\boldsymbol{\alpha}^+, \boldsymbol{\alpha}^- \ge \mathbf{0}$ such that $\boldsymbol{\alpha} = \boldsymbol{\alpha}^+ - \boldsymbol{\alpha}^-$, and obtain the following problem:

$$\min_{\boldsymbol{\alpha}^+, \boldsymbol{\alpha}^-} \frac{1}{2}(\boldsymbol{\alpha}^+ - \boldsymbol{\alpha}^-)^\top \boldsymbol{K}(\boldsymbol{\alpha}^+ - \boldsymbol{\alpha}^-) - (\boldsymbol{\alpha}^+ - \boldsymbol{\alpha}^-)^\top \boldsymbol{y} + \varepsilon\mathbf{1}^\top(\boldsymbol{\alpha}^+ + \boldsymbol{\alpha}^-), \quad \text{s.t. } 0 \le \alpha_i^+, \alpha_i^- \le \frac{1}{\lambda}. \tag{A.9}$$

It can be proven that the optimal solution to (A.8) and (A.9) satisfies $\alpha_i^+ = \max\{0, \alpha_i\}, \alpha_i^- = \max\{0, -\alpha_i\}$. Then the SVR problem becomes a quadratic programming problem that can be solved with DBCD-TR.

On the other hand, we provide a simpler strategy. Minimizing $J(\boldsymbol{\alpha}_{\mathcal{B}} + \boldsymbol{s})$ is equivalent to:

$$\min_{\boldsymbol{s} \in \Omega} \frac{1}{2}\boldsymbol{s}^\top \boldsymbol{K}_{\mathcal{B},\mathcal{B}}\boldsymbol{s} + \boldsymbol{s}^\top(\boldsymbol{K}_{\mathcal{B},:}\boldsymbol{\alpha} - \boldsymbol{y}_{\mathcal{B}}) + \varepsilon\|\boldsymbol{\alpha}_{\mathcal{B}} + \boldsymbol{s}\|_1, \quad \text{s.t. } \|\boldsymbol{s}\|_2 \le \Delta, \tag{A.10}$$

where $\Omega = \{\boldsymbol{s} : \|\boldsymbol{\alpha}_{\mathcal{B}} + \boldsymbol{s}\|_\infty \le \lambda^{-1}\}$ is the feasible set. Now we assume $\|\boldsymbol{\alpha}_{\mathcal{B}} + \boldsymbol{s}\|_1 \approx \|\boldsymbol{\alpha}_{\mathcal{B}}\|_1 + \text{sign}(\boldsymbol{\alpha}_{\mathcal{B}})^\top \boldsymbol{s}$. The equality holds when all elements $\boldsymbol{\alpha}_{\mathcal{B}}$ and $\boldsymbol{\alpha}_{\mathcal{B}} + \boldsymbol{s}$ have the same sign, and the errors occur on the different signed elements. Therefore, the trust region size should be sufficiently small to keep the sign consistency as much as possible. Then the trust region subproblem becomes the following:

$$\min_{\boldsymbol{s} \in \Omega} \frac{1}{2}\boldsymbol{s}^\top \boldsymbol{K}_{\mathcal{B},\mathcal{B}}\boldsymbol{s} + \boldsymbol{s}^\top(\boldsymbol{K}_{\mathcal{B},:}\boldsymbol{\alpha} - \boldsymbol{y}_{\mathcal{B}} + \text{sign}(\boldsymbol{\alpha}_{\mathcal{B}})), \quad \text{s.t. } \|\boldsymbol{s}\|_2 \le \Delta, \tag{A.11}$$

which falls back into the framework discussed in the main paper with $\boldsymbol{Q} = \boldsymbol{K}_{\mathcal{B},\mathcal{B}}, g = \boldsymbol{K}_{\mathcal{B},:}\boldsymbol{\alpha}_{\mathcal{B}} - \boldsymbol{y}_{\mathcal{B}} + \text{sign}(\boldsymbol{\alpha}_{\mathcal{B}})$.

## C. Details of Experiments

### C.1. Datasets

We evaluate the models with the datasets that are commonly used in the literature on kernel methods. In these datasets, MSD, SUSY, and HIGGS were used in (Rudi et al., 2017; Meanti et al., 2020), etc. The HEPC dataset was used in (Lin

et al., 2024), and the CIFAR-5M dataset was benchmarked in (Abedsoltan et al., 2023). Like most of the previous works, the z-score normalization is always applied to make data with zero mean and unit variance. The information on the datasets is summarized in Table A.1.

*Table A.1.* Summary of datasets used in experiments

| Dataset | Task | $n$ | $d$ | Data Split | Metrics |
|---------|------|-----|-----|------------|---------|
| MSD | Regression | $5.1 \times 10^5$ | 90 | 90% train, 10% test | Relative Error ($\downarrow$) |
| HEPC | Regression | $2.05 \times 10^6$ | 11 | 90% train, 10% test | RMSE ($\downarrow$) |
| SUSY | Binary Classification | $5.0 \times 10^6$ | 18 | 80% train, 20% test | AUC ($\uparrow$), Accuracy ($\uparrow$) |
| HIGGS | Binary Classification | $1.1 \times 10^7$ | 28 | 80% train, 20% test | AUC ($\uparrow$), Accuracy ($\uparrow$) |
| CIFAR-5M | 10-class Classification | $5.0 \times 10^6$ | 3072 | 80% train, 20% test | Accuracy ($\uparrow$) |

- Million-song dataset (MSD) (Bertin-Mahieux et al., 2011). This dataset contains audio features for year prediction. Available at `https://archive.ics.uci.edu/dataset/203/yearpredictionmsd`.

- Household Electric Power Consumption (HEPC) dataset: It is the same as the "HouseElec" dataset used in (Lin et al., 2024), available using the python package `https://github.com/treforevans/uci_datasets`.

- Supersymmetric particle classification (SUSY) dataset (Baldi et al., 2014): This is a binary classification task to distinguish between supersymmetric particles and background process. Available at `https://archive.ics.uci.edu/dataset/279/susy`.

- HIGGS dataset (Baldi et al., 2014): This is a binary classification task to distinguish between the Higgs boson and the background process. Available at `https://archive.ics.uci.edu/dataset/280/higgs`.

- CIFAR-5M dataset (Nakkiran et al., 2021): This is a generated dataset based on CIFAR-10. Available at `https://github.com/preetum/cifar5m`.

## C.2. Implementation details

We implement RFF of two widely-used kernels, the Gaussian and Laplacian, in inexact Joker. RFF is constructed by the following formulas:

$$\boldsymbol{\psi}(\boldsymbol{x}) = \sqrt{\frac{2}{M}} \cos(\boldsymbol{W}\boldsymbol{x} + \boldsymbol{b}), \tag{A.12}$$

where $\boldsymbol{W} \in \mathbb{R}^{M \times d}$ and $\boldsymbol{b} \in \mathbb{R}^M$ are random matrices. Each element of $\boldsymbol{b}$ is always sampled from the uniform distribution $U_{[0,2\pi]}$ independently. The distribution of $\boldsymbol{W}$ depends on the kernel:

- **Gaussian kernel:** Each element $w_{ij}$ is sampled independently from Gaussian distribution with zero mean and variance $\sigma^2$:

$$\mathcal{K}(\boldsymbol{x}, \boldsymbol{x}') = \exp\left(-\frac{\|\boldsymbol{x} - \boldsymbol{x}'\|^2}{2\sigma^2}\right), \quad p(w) = \mathcal{N}(0, \sigma^2), \tag{A.13}$$

- **Laplacian kernel:** Each element $w_{ij}$ is sampled independently from the Cauchy distribution with scale $1/\sigma$:

$$\mathcal{K}(\boldsymbol{x}, \boldsymbol{x}') = \exp\left(-\frac{\|\boldsymbol{x} - \boldsymbol{x}'\|_1}{\sigma}\right), \quad p(w) = \frac{\sigma}{\pi(w^2\sigma^2 + 1)}. \tag{A.14}$$

In our experiments, we implement five Joker models: KRR, Huber, SVC, SVR, and KLR. In the main paper, we merge SVC and SVR into Joker-SVM. The summary is shown in Table A.2.

Table A.2. Lookup table of Joker models in the experiments. $\mathsf{bEnt}(x) := x \log x + (1-x) \log(1-x)$ is the binary entropy function and $[\cdot]_+ := \max\{0, \cdot\}$.

| Model | Primal problem | Dual problem | Constraint |
|---|---|---|---|
| Joker-KRR | $\min_{\boldsymbol{\theta}} \frac{1}{2}\|\boldsymbol{\theta}\|^2 + \frac{1}{2\lambda}\sum_{i=1}^{n}(y_i - \langle\boldsymbol{\theta},\boldsymbol{\varphi}(\boldsymbol{x}_i)\rangle)^2$ | $\min_{\boldsymbol{\alpha}} \frac{1}{2}\boldsymbol{\alpha}^\top(\boldsymbol{K}+\lambda\boldsymbol{I})\boldsymbol{\alpha} - \boldsymbol{y}^\top\boldsymbol{\alpha}$ | $-\infty \leq \alpha_i \leq \infty$ |
| Joker-Huber | $\min_{\boldsymbol{\theta}} \frac{1}{2}\|\boldsymbol{\theta}\|^2 + \frac{1}{2\lambda}\sum_{i=1}^{n}(y_i - \langle\boldsymbol{\theta},\boldsymbol{\varphi}(\boldsymbol{x}_i)\rangle)^2$ | $\min_{\boldsymbol{\alpha}} \frac{1}{2}\boldsymbol{\alpha}^\top(\boldsymbol{K}+\lambda\boldsymbol{I})\boldsymbol{\alpha} - \boldsymbol{y}^\top\boldsymbol{\alpha}$ | $-\frac{\delta}{\lambda} \leq \alpha_i \leq \frac{\delta}{\lambda}$ |
| Joker-SVC | $\min_{\boldsymbol{\theta}} \frac{1}{2}\|\boldsymbol{\theta}\|^2 + \frac{1}{2\lambda}\sum_{i=1}^{n}[1 - y_i\langle\boldsymbol{\theta},\boldsymbol{\varphi}(\boldsymbol{x}_i)\rangle]_+^2$ | $\min_{\boldsymbol{\alpha}} \frac{1}{2}\boldsymbol{\alpha}^\top(\boldsymbol{K}+\lambda\boldsymbol{I})\boldsymbol{\alpha} - \boldsymbol{y}^\top\boldsymbol{\alpha}$ | $0 \leq \alpha_i y_i \leq \infty$ |
| Joker-SVR | $\min_{\boldsymbol{\theta}} \frac{1}{2}\|\boldsymbol{\theta}\|^2 + \frac{1}{\lambda}\sum_{i=1}^{n}[|y_i - \langle\boldsymbol{\theta},\boldsymbol{\varphi}(\boldsymbol{x}_i)\rangle| - \varepsilon]_+$ | $\min_{\boldsymbol{\alpha}} \frac{1}{2}\boldsymbol{\alpha}^\top\boldsymbol{K}\boldsymbol{\alpha} + \varepsilon\|\boldsymbol{\alpha}\|_1 - \boldsymbol{y}^\top\boldsymbol{\alpha}$ | $-\frac{1}{\lambda} \leq \alpha_i \leq \frac{1}{\lambda}$ |
| Joker-KLR | $\min_{\boldsymbol{\theta}} \frac{1}{2}\|\boldsymbol{\theta}\|^2 + \frac{1}{\lambda}\sum_{i=1}^{n}\log(1 + e^{-y_i\langle\boldsymbol{\theta},\boldsymbol{\varphi}(\boldsymbol{x}_i)\rangle})$ | $\min_{\boldsymbol{\alpha}} \frac{1}{2}\boldsymbol{\alpha}^\top\boldsymbol{K}\boldsymbol{\alpha} + \frac{1}{\lambda}\sum_{i=1}^{n}\mathsf{bEnt}(-\lambda\alpha_i y_i)$ | $0 \leq \alpha_i y_i \leq \frac{1}{\lambda}$ |

## C.3. Parameter settings

The major hyperparameters of the models are shown in the experiments in Table A.3. We tend to use the best parameters reported in literature. Median heuristic is used to set the bandwidth parameter $\sigma$ of kernels, denoted as "median" in Table A.3. Our empirical results show that it always produces satisfying performance, and its rationality is also supported by (Garreau et al., 2018).

*Table A.3.* The major hyperparameters of the models in the experiments. "NA" means not applicable.

| Methods | Parameters | Dataset | | | | |
|---|---|---|---|---|---|---|
| | | MSD | HEPC | SUSY | HIGGS | CIFAR-5M |
| Falkon | $\lambda_{\mathrm{fal}}$ | $10^{-6}$ | $10^{-9}$ | $10^{-6}$ | $10^{-8}$ | $10^{-8}$ |
| | $M$ | 25000 | 25000 | 10000 | 25000 | 20000 |
| | precision | float32 | float64 | float32 | float32 | float32 |
| | kernel, $\sigma$ | Gaussian, 6 | Gaussian, 4 | Gaussian, 4 | Gaussian, 5 | Gaussian, median |
| | epochs | 20 | 50 | 20 | 10 | 50 |
| LogFalkon | $\lambda_{\mathrm{lgf}}$ | | | $10^{-9}$ | $10^{-9}$ | |
| | $M$ | | | 25000 | 25000 | |
| | #Newton step | NA | NA | 8 | 8 | NA |
| | kernel, $\sigma$ | | | float32 | float32 | |
| | epochs | | | 15 | 15 | |
| EigenPro3 | $M$ | 80000 | 80000 | 50000 | $10^5$ | $10^5$ |
| | kernel, $\sigma$ | Laplacian, median | Laplacian, median | Laplacian, median | Laplacian, median | Laplacian, median |
| | epochs | 30 | 50 | 50 | 30 | 50 |
| ThunderSVM | $c$ | 1 | 16 | 8 | 32 | 32 |
| | kernel, $\sigma$ | Gaussian, median | Gaussian, median | Gaussian, median | Gaussian, median | Gaussian, median |
| | $\epsilon$ in SVR | 0.25 | 0.25 | NA | NA | NA |
| Joker-KRR | $\lambda$ | 1 | $2^{-7}$ | $2^{-5}$ | $2^{-7}$ | $2^{-7}$ |
| | $M$ | Exact model | 50000 | $10^5$ | $10^5$ | $2 \times 10^5$ |
| | kernel, $\sigma$ | Gaussian, median | Laplacian, median | Laplacian, median | Laplacian, median | Gaussian, median |
| | block size | 2048 | 512 | 512 | 512 | 512 |
| | #iterations | 10000 | 25000 | 10000 | 50000 | 40000 |
| Joker-Huber | $\lambda$ | 1 | $2^{-7}$ | $2^{-5}$ | $2^{-7}$ | $2^{-7}$ |
| | $M$ | Exact model | 50000 | $10^5$ | $10^5$ | $2 \times 10^5$ |
| | $\delta$ | 2 | 1 | 1 | 1 | 1 |
| | kernel, $\sigma$ | Gaussian, median | Laplacian, median | Laplacian, median | Laplacian, median | Gaussian, median |
| | block size | 2048 | 512 | 512 | 512 | 512 |
| | #iterations | 10000 | 25000 | 10000 | 50000 | 40000 |
| Joker-SVM | $\lambda$ | 1 | $2^{-7}$ | $2^{-5}$ | $2^{-7}$ | $2^{-7}$ |
| | $M$ | Exact model | 10000 | $10^5$ | $10^5$ | $2 \times 10^5$ |
| | kernel, $\sigma$ | Gaussian, median | Laplacian, median | Laplacian, median | Laplacian, median | Gaussian, median |
| | $\epsilon$ in SVR | 0.25 | 0.25 | NA | NA | NA |
| | block size | 2048 | 512 | 512 | 512 | 512 |
| | #iterations | 10000 | 25000 | 10000 | 50000 | 40000 |
| Joker-KLR | $\lambda$ | | | 2 | $2^{-3}$ | $2^{-7}$ |
| | $M$ | | | $10^5$ | $10^5$ | $2 \times 10^5$ |
| | kernel, $\sigma$ | NA | NA | Laplacian, median | Laplacian, median | Gaussian, median |
| | block size | | | 1024 | 1024 | 1024 |
| | #iterations | | | 40000 | 50000 | 50000 |

