# OpenReview forum: "Joker: Joint Optimization Framework for Lightweight Kernel Machines"
_ICML.cc/2025/Conference — ICML 2025 poster_

### Official Review · Reviewer_aGkm · 2025-03-06

**Overall Recommendation:** 3

**Summary:**

This paper proposes a novel algorithm for extremely large-scale kernel machines. Its main contribution lies in Theorem 2.1, which reformulates the objective function in the dual problem of kernel methods into a form based on decoupled conjugate functions. This ensures convexity and strong duality, making it possible to use the block coordinate descent method for optimization. In each subproblem, the conjugate function is further approximated by performing a Taylor expansion and optimizing the approximation. Experimental results show that this method outperforms some previous algorithms for very large-scale kernel machines.

**Claims And Evidence:**

Strength:

1. The formulation provided in Theorem 2.1 is ingenious. The subsequent ideas, including block decomposition optimization, approximate optimization, and the use of random features for approximation, are sound.

Weakness:

1. I believe the author's definition of "exact" is somewhat biased. The core of the proposed method is to focus only on one subset during each optimization step (as in Equation (6)). For this subproblem, approximate optimization is required (as in Equation (7)). Essentially, this is similar to methods like Nystrom, which select a sub-dataset and are inherently inexact. If the authors wish to claim that their method is "exact", they need to theoretically guarantee that the iterative optimization of the subproblem with box constraints can reach the optimal solution of Problem (4) or provide some error bounds. The authors could refer to related proofs in block coordinate descent methods to explain this.

2. In the experiments, the authors discussed the impact of the choice of $|\mathcal{B}|$ on the results, but I did not notice some discussion regarding the setting of other parameters, such as the maximum region size $\Delta$.

**Essential References Not Discussed:**

None.

**Experimental Designs Or Analyses:**

Good.

**Methods And Evaluation Criteria:**

The compared methods and criteria are reasonable.

**Other Comments Or Suggestions:**

1. In Equation (1), it should be written as $\langle \theta, \phi(x_i) \rangle_{\mathcal{H}}$.

2. In Equation (A.2), $w$ should be replaced with $\theta$. Additionally, to align with Equations (A.3) and (A.4), it would be better to use $\alpha_i(u_i - \theta^\top \phi(x_i))$.

3. Due to the notation $\xi_y(u)$ in the main context, it is preferable to use something like $(f \square g)(u) \coloneqq \inf_p f(p) + g(u - p)$.

**Other Strengths And Weaknesses:**

See Claims And Evidence.

**Questions For Authors:**

None.

**Relation To Broader Scientific Literature:**

None.

**Theoretical Claims:**

There are some typos in the proof of Theorem 2.1. Please refer to "Other Comments or Suggestions".

---

> ### Author Rebuttal · Authors · 2025-03-31
>
> ## Weak. 1: "Exact" and "Inexact"
> In the context of this paper, the "inexact" and "exact" are not about the approximation of eq.(7),
> which has been stated in Section 1.2.
> "Exact" refers to solving the problem eq.(1) without approximating the kernel function $K(\cdot,\cdot)$, and these methods usually involves $n$-dimensional variable $\alpha$.
> "Inexact" approximates the kernel function with a map $\psi(\cdot)$ (e.g., Nystrom and RFF) such that $K(x,x')\approx \psi(x)^\top\psi(x')$ and reduces computation burden.
> These terminologies are consistent with [1].
>
> Besides, the trust region method in solving eq.(6) is not similar to Nystrom.
> As a well-studied optimization technique,
> the trust region method has a theoretical guarantee of convergence to the optimal point.
> We have explained it in response to Q3 of reviewer MsAQ.
>
> ## Weak. 2: impact of other hyperparameters
> Thank you for the suggestion.
> We have listed the optimal hyperparameters in Table A.3.
> The block size is emphatically discussed because it directly influences the memory footprint and the convergence,
> while the max region size $\Delta_{max}$ has a lower impact.
> This is because the trust region procedure will adaptively tune the region size to ensure a sufficient decrease of the objective function.
> So the final performance is usually insensitive to $\Delta_{max}$ as long as it is not extremely tiny or huge.
> Empirically, $\Delta_{max}\in[4,64]$ is moderate.
>
>
> # Other comments
> Thank you for your careful review.
> We have corrected the typos you mentioned.
>
> # Ref.
> [1] Rahimi et al. Random Features for Large-Scale Kernel Machines, Neurips2007.

---

> > ### Comment · Reviewer_aGkm · 2025-04-02
> >
> > Thanks for your rebuttal. My concerns are addressed. I decided to raise my score to 3.

---

> > > ### Author Response · Authors · 2025-04-02
> > >
> > > Thank you for your feedback. We appreciate your approval of our work.

---

### Official Review · Reviewer_Zf7v · 2025-03-11

**Overall Recommendation:** 3

**Summary:**

The authors propose a general optimization scheme for kernel machines. Similar to Teo et al. (2009), conjugate loss functions allow for unified representations that are solved with block coordinate descent in dual space (Sorensen, 1982). The authors report on empirical results showing that their optimizer is more efficient than its competitors.

[Slightly updated score after rebuttal]

**Claims And Evidence:**

The authors claim generality of their approach due to the use of conjugate loss functions. The idea is not novel and always lead to unified representations of quadratic objectives (e.g., regularized empirical risks). The empirical results show that the present approach is actually saving a great deal of memory when compared with other optimizers. I am not up to date in this domain anymore but back than the fastest solver was OCAS and a similar versatility were contained in Teo et al. (see below for references). Unfortunately, both are not cited or compared in the paper.

I checked the proof of Theorem 2.1. The proof seems to be correct, but may suffer from some smaller inaccuracies.
From my understanding, the last sum in Equation A.2. (line 616/617) should be :
$$ \sum_{i=1}^{n}\alpha_i(u_i-\theta^T\varphi(x_i))$$
changing the sign of the expression to be coherent with Equations A.3 and A.4, and replacing the $\omega$ with  $\theta$.

**Essential References Not Discussed:**

V. Franc and S. Sonnenburg. Optimized cutting plane algorithm for support vector machines. In A. McCallum and S. Roweis, editors, Proceedings of the International Conference on Machine Learning, pages 320–327. Omnipress, 2008. (OCAS used to be the fastest SVM solver back then)

C. H. Teo, S.V.N. Vishwanathan, A. Smola, Q. Le: Bundle Methods for Regularized Risk Minimization. Journal of Machine Learning Research 11 (2010) 311-365, 2009.

**Experimental Designs Or Analyses:**

See above.

**Methods And Evaluation Criteria:**

The approach is evaluated against four baselines (two are missing in my view, see additional literature below) on five large data sets. The proposed method appears to beat the baselines in terms of memory footprint and often also predictive accuracies.

**Other Comments Or Suggestions:**

In my PDF, the infimal convolution is denoted by a square that seems identical to the end of a proof environment. Is that they symbol you wanted us to see? The authors claim that "Falkon based methods are the fastest, but the gap between Joker and them is not substantial" (line 370-373). This seems to be a rather subjective interpretation of the results in Table 4, where Falkon performs comparatively at only about 1/6 of the time requirement on the MSD dataset and LogFalkon achieving the best results at about half the time of the best Joker based approach. There seems to be a typo in Equation A.5, where it should be  $\alpha_i\alpha_j$ instead of $\alpha_i\alpha_k$.

**Other Strengths And Weaknesses:**

The proposed method is actually very memory efficient and performs well compared with the baselines. There is a lack of theory (eg convergence) for such a paper in my view.

**Questions For Authors:**

What about convergence proofs or bounds? How does performance and efficiency compare to OCAS? How does the approach relate to Teo et al?

**Relation To Broader Scientific Literature:**

The paper seems a bit outdated although faster computation is always appreciated, even for kernel machines that are kind of outruled by networks at the moment. There has been quite a great deal of optimization approaches for kernel machines end of the nineties until perhaps 2010. I am citing two papers that came to my mind but I have to admit that I don't remember much from what I know once...

**Theoretical Claims:**

The authors show hardly any theoretical claims. It would have been nice to learn more about convergence of the inexact variant with randomization of features.

---

> ### Author Rebuttal · Authors · 2025-03-31
>
> # Questions
> ## Q1. Convergence
> The linear convergence rate of the proposed DBCD-TR can be proven using Polyak-Lojasiewicz condition.
> However, it is a loose bound and does not highlight the advantage of DBCD-TR.
> The tighter bound (a reasonable guess is a superlinear rate) is a challenging work, considering that DBCD-TR itself is a quite complicated algorithm.
> Therefore, we tend not to present the convergence result in this paper, and focus on the practical design of the algorithm.
> We have explained this in response to Q5 of the reviewer MsAQ.
>
> ## Q2. Comparison on OCAS [1]
> We argue that OCAS and the BMRM are unsuitable to be benchmarks in our experiment for the following reasons.
>
> 1. They are designed for linear models but not kernel methods.
> This distinction is important because kernel methods are usually associated with an ill-conditioned problem,
> which means that *a fast solver for a linear model could be slow in the kernel regime*.
> The recent kernel methods, including EigenPro series, Falkon, LogFalkon aim to overcome the ill-conditioning via preconditioning or Newton methods.
> Our solution, DBCD-TR, uses **trust region** method to incorporate truncated Newton step and optimizes **multiple variables (a block)** at a time to leverage the merit of parallel computing.
>
> 2. They are outdated since LIBLINEAR is faster than them, as shown in the experiments of [3].
> Theoretically, LIBLINEAR is based on dual coordinate descent [2] and has a linear convergence rate $O(\log(1/\epsilon))$ superior to $O(1/\epsilon)$ of OCAS.
> So if we must make a comparison, we should choose LIBLINEAR but not [2] and [3].
>
> 3. The proposed DBCD-TR is much faster than LIBLINEAR.
> As mentioned before, OCAS, BMRM and LIBLINEAR are for linear models.
> But it does not mean that they cannot handle the kernel models.
> As mentioned by [3], we can first obtain the random Fourier feature (RFF) of data and apply the linear models,
> thus obtaining an inexact kernel machine.
> Using this approach, we run LIBLINEAR (GPU implementation for fairness) on HIGGS ($M=10^5$, same as Joker-SVM in our experiment) during rebuttal.
> However, it hardly converges and only obtains an accuracy of 64.7% after 3 days.
> For OCAS and BMRM, they will cost more time.
> Therefore, we think they are unsuitable to be benchmarks in our experiment.
>
> ## Q3. Relation to BMRM [3]
> BMRM is also for linear SVM as OCAS and LIBLINEAR.
> BMRM minimizes the piecewise lower bound (PLB) of the primal objective via the Fenchel conjugate.
> Then it minimizes PLB by optimizing a series of dual problems.
> In contrast, Joker does not approximate the primal objective function,
> but optimizes the dual objective directly.
> We have presented more discussion in Q2.
>
> # Other Comments
> ## 1. Typos in Appendix
> We have fixed the typos you mentioned.
> Thank you for pointing them out.
>
> ## 2. Notation of infimal convolution
> Yes, we note the infimal convolution as "$\square$" following the notation of most textbooks.
>
> ## 3. Interpretation of Table 4
> Thank you for your comments.
> Our words of "substantial" may be imprecise.
> What we want to state is that the time gap between other methods like EigenPro and Falkon is large,
> and Joker alleviate this gap significantly.
> We realize that the analysis combined with data may be convincing:
> EigenPro3 and ThunderSVM use at least 10x training time than Falkon on MSD,
> and Joker reduces it to 5~6x time.
> Especially on HIGGS, EigenPro3 needs 36x time (18 hours) of Falkon (0.5 hour),
> and Joker reduces the gap to 2x time (1 hour).
>
> # Ref.
> [1] Franc et al. Optimized cutting plane algorithm for support vector machines. ICML 2008.
>
> [2] Hsieh et al. A dual coordinate descent method for large-scale linear SVM. ICML 2008.
>
> [3] Teo et al. Bundle Methods for Regularized Risk Minimization. JMLR, 2009.

---

> > ### Comment · Reviewer_Zf7v · 2025-04-02
> >
> > Thanks!

---

> > > ### Author Response · Authors · 2025-04-02
> > >
> > > Thank you for your response. We are not certain whether your concerns are well addressed. We would like to hear detailed feedback from you for further discussion.

---

### Official Review · Reviewer_MsAQ · 2025-03-12

**Overall Recommendation:** 3

**Summary:**

This paper explores a joint optimization framework for diverse kernel models, including KRR, logistic regression, and support vector machines.
The authors employed a dual block coordinate descent method with trust region (DBCD-TR) and kernel approximation with randomized features to solve the proposed model, which makes the algorithms have low memory costs and high efficiency in large-scale learning.

**Claims And Evidence:**

The proposed approach shows a good performance on some tasks.

**Essential References Not Discussed:**

The paper contains sufficient discussed references

**Experimental Designs Or Analyses:**

Experimental results demonstrate a good performance compared to baselines.

**Methods And Evaluation Criteria:**

The evaluation metric used in this paper is reasonable.

**Other Comments Or Suggestions:**

No suggestions.

**Other Strengths And Weaknesses:**

This paper explores a joint optimization framework for diverse kernel models, including KRR, logistic regression, and support vector machines. For weaknesses, please refer to the problem below.

**Questions For Authors:**

1)Classical kernel-based methods usually contain biases. Why are the biases not considered in the proposed model?

2)For Eq.(6), one may employ projected gradient methods to solve them. What is the benefit of the rust-region method adopted in this article?

3)The authors employ Taylor expansions in Eq.(7). This approximation will introduce errors such that the solution deviates from the optimal solution.

4)When nonconvex loss functions are employed, there exists the dual gap between the primal problem and the dual problem. How to deal with this problem?

5)It would be much better to prove the convergence of the algorithm under proper assumptions.

**Relation To Broader Scientific Literature:**

The proposed techniques can be employed to deal with various kernel models beyond KRR, thereby giving a more comprehensive solution for kernel learning and related applications.

**Theoretical Claims:**

I checked the proofs of  some theorems, and they sound correct.

---

> ### Author Rebuttal · Authors · 2025-03-31
>
> ## Q1. bias term.
> It is for simplicity and keeping consistency with the recent literature of kernel methods, where they also do not consider it.
> A simple way to include the bias is to append a constant after $\varphi(x)$.
> However, adding this term generally has no impact on the final performance,
> so our paper as well as the related work tends to disregard it.
> This is good feedback, and we will briefly explain it in our paper.
>
> ## Q2. benefit of trust region.
> The main reason is that projected gradient descent (PGD) converges slowly.
> The best rate of PGD is linear, specifically, $O(\kappa\log(1/\epsilon))$, where $\kappa$ is the condition number.
> But due to the kernel matrix, eq.(6) usually has large $\kappa$ and the linear rate may be insufficient.
> So many latest kernel methods, such as [1] use second-order information for acceleration.
> The proposed trust region (TR) has the same purpose and can reach a superlinear rate [2],
> significantly faster than PGD.
>
> ## Q3. approximation of eq.(7).
> This is a misunderstanding of TR method.
> It aims to find multiple steps toward the optimal point,
> and each point is restricted in a small region (i.e., TR) wherethe  approximation error of eq.(7) is tiny.
> The procedure of TR method guarantees that the steps always decrease the objective function and finally reach the optimality (especially, with superlinear convergence rate [2]).
> If the approximation error of eq.(7) is large and yields a suboptimal step, TR's procedure will reject such a step automatically and decrease the region size.
> It may be clearer to see Algorithm 3 in Appendix B.
>
> ## Q4. Nonconvex loss.
> Our paper focuses on the large-scale kernel method with the convex losses, but not the nonconvex ones.
> We have stated that "Joker focuses on convex problem..." (line 143),
> and nonconvexity is not within the scope of our paper.
> Despite this, Joker still significantly develops kernel models with diverse loss functions,
> and improves the speed and memory footprint compared to the existing kernel methods.
>
> Kernel methods with nonconvex losses should be a future work.
> The approaches may be significantly different from the proposed Joker.
> The feasible approaches may include Difference of Convex functions (DC) programming, convex relaxation, and approximation
> (e.g., piecewise linear approximation method [1] mentioned by reviewer Zf7v).
>
> ## Q5. Convergence.
> We decided not to include convergence for the following reasons:
>
> 1. Our work is more practical, as stated by reviewer 1dCw, it has "the practical nature".
> The proposed method has remarkable contributions:
> largely reducing the memory footprint of large-scale kernel methods, and unifying the optimization scheme of different kernel models.
> We believe the two advances are significant enough in practice.
>
> 2. We feel that the rigorous proof of the convergence is non-trivial and deserves in-depth study in future work.
> The proposed DBCD-TR algorithm is a relatively complicated algorithm.
> Block coordinate descent and the truncated CG-Steihaug method (Algorithm 1) may become two major challenges in investigating the convergence rate.
> Indeed, a simple linear convergence rate $O(n\log(1/\epsilon)/|B|)$ can be obtained easily using the Polyak-Lojasiewicz condition.
> However, it is vacuous since the first-order block coordinate descent methods also have this rate.
> So the linear rate cannot highlight the improvement of DBCD-TR.
> Considering the usage of second-order information, it is reasonable to guess that DBCD-TR can have a superlinear convergence rate, as hinted in Section 6 of [2].
>
> Nonetheless, we still can give an outline of the convergence proof:
> - Sufficient descent. We first prove that truncated CG-Steihaug satisfies the sufficient descent condition in [3].
> - Global Convergence to a stationary point. The limit point of the proposed trust region is a stationary point.
> - Local Superlinear convergence: The trust region process (Algorithm 2) converges with a superlinear rate same as the projected Newton.
> - Giving the iteration complexity using the above results, and investigate the influence of the Hessian in different cases.
>
> The above illustrates the convergence of one block eq.(6).
> For the outer loop (the block coordinate),
> we may establish the iteration complexity result following the analysis in [4] and combining the decrease made by the trust region.
>
> # Ref.
> [1] Teo et al. Bundle Methods for Regularized Risk Minimization. JMLR, 2009.
>
> [2] Nutini et al. Let's Make Block Coordinate Descent Converge Faster: Faster Greedy Rules, Message-Passing, Active-Set Complexity, and Superlinear Convergence. JMLR, 2022.
>
> [3] Baraldi et al. Efficient proximal subproblem solvers for a nonsmooth trust-region method. Computational Optimization and Applications, 2025.
>
> [3] Li et al. On Faster Convergence of Cyclic Block Coordinate Descent-type Methods for Strongly Convex Minimization. JMLR, 2018.

---

> > ### Comment · Reviewer_MsAQ · 2025-04-07
> >
> > Thank you for your detailed answer. I would like to raise the score to 3.

---

> > > ### Author Response · Authors · 2025-04-08
> > >
> > > Thank you for raising your score to 3, which is encouraging feedback! We appreciate your recognition of our work.

---

### Official Review · Reviewer_1dCw · 2025-03-18

**Overall Recommendation:** 3

**Summary:**

The paper proposes Joker, a novel optimization scheme that aims at scaling kernel methods beyond current computational limitations. It is versatile and can handle several objective functions in a similar manner. The core idea of Joker is to solve the dual problem with a block coordinate descent with trust region. An approximate version based on random fourier features is also developed; its time and space complexity are low and Joker exhibits good performance on large scale (for kernel methods) datasets.

**Claims And Evidence:**

Claims of better or even performance under smaller memory budget are convincing.

**Essential References Not Discussed:**

None.

**Experimental Designs Or Analyses:**

The experimental designs are correct. However no code is provided, and given the practical nature of the paper, I would have liked to audit the code.

**Methods And Evaluation Criteria:**

Yes, the evaluation criteria makes sense.

**Other Comments Or Suggestions:**

- [Line 94] A Mercer kernel is typically a kernel that satisfies the assumptions needed to apply Mercer's theorem: continuous on a compact domain. Any positive definite kernel uniquely induces a RKHS.

- [Line 105] Typo $\phi(x_i)_{\mathcal{H}}$

- [Line 107] where $\phi(x)$ is the linear map -> a linear map, as there can be several.

- [Line 132] the closeness and the convexity

- [Line 171] "It implies that if the constraints are properly handled,
kernel Huber regression can be as efficient as KRR" -> what this implies is that if the constraints are satisfied by the dual solution to the KRR, then both solutions coincide. There exist cases of contaminated data where the Huber loss estimator outperforms the KRR precisely because the dual coefficients must pertain to a smaller ball and cannot be too influenced by outliers.

- [Line 189] "in some rare cases" -> either $f$ is assumed twice differentiable, or it is not. Pick one but do not claim full generality while proposing ad-hoc modifications.

- [Line 230] "In our implementation, $T_{TR} \leq 50$" -> ok but that does not mean that $n$ does not appear in the complexity. For example if you use blocks of size $512$ and have $512k$ samples you would approximately need to go through 1k times to make a pass on all the dual variables. And this would not even guarantee convergence. Or am I wrong here ?

- [Line 254] "we obtain $\theta = \sum_{i=1}^n \alpha_i \psi(x_i)$" -> usually when using random features, one big advantage is that the parameter space is reduced, so that $\theta$ can be directly searched for on the feature space. Could you comment on the difference with your approach here ?

- [Line 272] The results from [Rahimi and Recht, 2007] are known to be suboptimal and certainly do not represent a theoretical guide for setting $M$. See e.g. [Error Bounds for Learning with Vector-Valued Random Features, Lanthaler and Nelsen, Neurips 2023].

**Other Strengths And Weaknesses:**

+: The paper is very relevant to ICML and the practical improvement over existing methods is valuable.

-: The writing is often not precise, see comments or suggestions box.

-: Inexact Joker relies on random fourier features, thus is limited to shift-invariant kernel (as proposed at least).

**Questions For Authors:**

My main criticism about the paper is that I find that it lacks rigor. Improving on the points raised in "other comments or suggestions" is critical to me.

**Relation To Broader Scientific Literature:**

- The Eigenpro series of papers could be better discussed in the introduction.

- The most recent addition to large scale KRR might be the arxiv paper [1], from July 2024, with a revised version from February 2025. However, given that it is recent and likely not through with the review process itself, I understand that it might be touchy to ask for a comparison. Still, it deserves more than the current hard-to-notice citation.

Otherwise the related work section is well organized. The idea of using trust regions on the dual problem is novel and promising, especially the part about the quadratic extension at (8).

The contribution from theorem 2.1 is minimal. The dualization of the regularized empirical risk minimization problem in a RKHS has been extensively studied over two decades. Similarly, the contribution of proposition 2.2 is again not original, it is a straightforward extension of the case of the infimal convolution of two loss functions.

[1] Have ASkotch: A Neat Solution for Large-scale Kernel Ridge Regression, Rathore et al.

**Theoretical Claims:**

Theoretical claims from section 2 are correct.

---

> ### Author Rebuttal · Authors · 2025-03-31
>
> ## Weak. 1: precise writing.
> We appreciate your suggestions to make the expression more precise.
> The following are our responses and the plan of revision.
>
> (line 94): Your expression is rigorous. We should add that $K$ satisfying Mercer condition is positive definite (Mercer condition is equivalent to positive-definiteness).
>
> (line 105, 107): Thank you for pointing out the typos. We have fixed them.
>
> (line 132): Yes it should be convexity. We have corrected it.
>
> (line 171): Your understanding on robust regression and KRR are precise.
> However, what we want to emphasize is the efficiency, i.e.,
> we can solve kernel Huber regression and KRR within a similar time.
> We rewrite this sentence to make it clearer.
>
> (line 189): Thank you. We noticed that it is imprecise expression.
> We move this sentence after we finish the discussion on the twice-differentiable $f$.
>
> (line 230):
> You may misunderstand $T_{TR}$.
> In fact, $T_{TR}$ is the iteration number of trust region procedure in solving eq.(6)(*maxIter* in Algorithm 3),
> which is small because the trust region method converges fast (typically superlinear).
>
> The number of block update iterations ($T$ in Algorithm 2) is large, typically many times of $n/|B|$.
> In your example with 512000 samples and a block size of 512, needing 1000 times to go through all data (one pass),
> we may set $T=10000$ and exactly go through 10 passes.
> Another example in our experiments, we use $T=50000$ with block size 512 for Joker-SVM on HIGGS (see Table A.3), roughly complete $40000*512/5M\approx 4$ passes.
>
> We clarify three iteration procedures:
> - Algorithm 2: DBCD-TR outer-loop, iteration times $T\sim O(n/|B|)$.
> - Algorithm 3 (in Appendix B): called by Algorithm 2, iteration times $T_{TR}\leq50$.
> - Algorithm 1: called by Algorithm 3, iteration times $T_{CG}\leq10$.
>
> (line 254): We regard "directly searched for on the feature space" as optimizing eq.(2) directly using the approximation feature $\varphi(x):=\psi(x)$.
> This is related to the primal-based methods involving $M$ variables and can be related to [1] and Falkon (using Nystrom feature).
> To obtain a promising convergence rate these methods usually utilize second-order algorithms.
> The challenges occurs when handling the Hessian or precondition matrix that may cause $O(M^2)$ computation.
> In contrast, the proposed Joker optimizes the dual variables but not eq.(2) itself.
> we only maintains the KKT condition $\theta=\sum_{i=1}^n\alpha_i\psi(x_i)$ once the dual variables updated using eq.(11).
> This procedure aims to reduce the computation complexity of $K_{B,:}\alpha_B$, as stated in Section 2.3.
> Compared to the direct method,
> the proposed dual optimization leverages the separable structure of eq.(4),
> allowing the block coordinate descent to process a small working set in each iteration.
> In this way, the computation of processing Hessian is reduced to $O(|B|^2)$.
>
> (line 272): We sincerely thank your suggestion.
> The provided reference proves that a generalization error of $O(n^{-1/2})$ can be obtained using $O(n^{1/2})$-dimensional RFF.
> This can be more useful in practice to guide the selection of $M$.
> We will update the our citation.
> ## Weak. 2: limitation of RFF?
> RFF is proposed for shift-invariant kernels initially.
> However, it has been developed to diverse kernels like dot-product kernels and additive kernels.
> Table 1 of [2] gives a good summary.
> Fastfood [3] also presents a principled way to construct random features beyond shift-invariant kernels.
> Regarding NTKs, [4] also provides a fast algorithm to obtain their explicit features.
> These results allow for obtaining RFF of a broad range of kernels and it is no longer a limitation of Joker.
>
> # Other
> ## Access to Code.
> Response to "However no code is provided...":
> We actually provided the code.
> The anonymous GitHub link is at the bottom of page 6 of the PDF.
> We welcome you to review the code and give suggestions.
> ## Related work.
> Thank you for your approval and suggestions on related work.
> Indeed, we noticed that ASkotch [5] was updated after our submission.
> We found many new results, and we will update our review of this paper.
> ## Theorem 2.1
> Indeed, many results similar to Theorem 2.1 was mentioned in the literature.
> We did not list Theorem 2.1 as our contribution.
> This theorem aims to highlight the dual formulation of the kernel methods,
> which is the key problem of this paper.
> # Ref.
> [1] Hsia et al. Preconditioned conjugate gradient methods in truncated Newton frameworks for large-scale linear classification, ACML2018.
>
> [2] Dai et al. Scalable Kernel Methods via Doubly Stochastic Gradient, Neurips2014, latest report: arxiv.org/pdf/1407.5599.
>
> [3] Le et al. Fastfood: Approximate Kernel Expansions in Loglinear Time, ICML2013.
>
> [4] Han et al. Fast Neural Kernel Embeddings for General Activations, Neurips2022.
>
> [5] Rathore et al. Have ASkotch: A Neat Solution for Large-scale Kernel Ridge Regression, arxiv.

---

> > ### Comment · Reviewer_1dCw · 2025-04-07
> >
> > I would like to thank the authors for their detailed answer, especially about the number of iterations and the code.
> >
> > I am very impressed with the practical performances shown in the paper, but have mixed feeling about the lack of guarantees (I know that it is a lot of work to get them).
> >
> > I'm still raising my score a bit. I'm leaning towards acceptance.

---

> > > ### Author Response · Authors · 2025-04-07
> > >
> > > We appreciate your approval of our work!
> > >
> > > Indeed, the theoretical results (e.g., convergence and generalization) of DBCD-TR are complicated and may not be suitable to be presented clearly in such a single work, so we are thankful for your understanding.
> > > We are making great efforts to study the theoretical guarantee thoroughly and hope to present it in the future.

---

### Decision · Program_Chairs · 2025-05-01

**Decision:**

Accept (poster)

**Comment:**

This paper addresses the scalability of kernel methods by proposing a unified optimization framework for various kernel models—including KRR, logistic regression, and support vector machines.  Joker uses a dual block coordinate descent method with trust region and randomized feature-based kernel approximation. It significantly reduces memory usage (up to 90%) while maintaining or surpassing the training speed and performance of state-of-the-art methods.

The paper is well written, and the empirical performance is impressive. The convergence proof is lacking, but the authors gave a reasonable outline of the proof in the rebuttal. Overall, the paper is a good addition to the conference.